# IGF1R-phosphorylated PYCR1 facilitates ELK4 transcriptional activity and sustains tumor growth under hypoxia

Ke Zheng[1,2,5], Nannan Sha[2,5], Guofang Hou[2], Zhuyun Leng[3], Qin Zhao[2], Li Zhang[4], Lingnan He[3], Meidong Xu[3] ✉, Yuhui Jiang[2] ✉ & Tao Chen[3] ✉

The proline synthesis is importantly involved in tumor growth under hypoxia, while the underlying mechanism remains to be further investigated. Here we show that pyrroline-5-carpoxylate reductase-1 (PYCR1), displaying a constant nuclear localization, is phosphorylated by nuclear IGF1R at Tyrosine 135 under hypoxia; this phosphorylation promotes the binding of PYCR1 to ELK4 and thus PYCR1 recruitment to ELK4-targeted genes promoter. Under hypoxia, ELK4-binding ability and enzymatic activity of PYCR1 are both required for ELK4-Sirt7-mediated transcriptional repression and cell growth maintenance, in which PYCR1-catalyzed NAD$^+$ production stimulates the deacetylation activity of Sirt7 on H3K18ac that restrains genes transcription. Functionally, PYCR1 Tyr-135 phosphorylation exerts supportive effect on tumor growth under hypoxia, and the level of PYCR1 Tyr-135 phosphorylation is associated with malignancy of colorectal cancer (CRC). These data uncover the relationship between the compartmentally metabolic activity of PYCR1 and genes transcription regulation, and highlight the oncogenic role of PYCR1 during CRC development.

Proline synthesis from glutamate is sequentially catalyzed by P5C synthase (P5CS/ALDH18A1) and pyrroline 5-carboxylate reductases (PYCR1, PYCR2 PYCR3/L), both generation of the intermediate glutamate 5-semialdehyde (GSA), which spontaneously cyclizes to form pyrroline 5-carboxylate (P5C), and eventual production involves oxidation of NAD(P)H and thus is linked with regulation of redox balance. Overexpression of PYCR1 is prevalently detected in various types of cancer and is importantly implicated in tumor development[1,2]. Recent studies indicate proline synthesis can be circumstantially enhanced under conditions that are unbeneficial for redox homeostasis or lead to a limitation for tricarboxylic acid (TCA) cycle activity[3,4]. In accordance, it is reported that PYCR1 activity plays an indispensable role in the TCA cycle sustaining under hypoxia in an NADH oxidation-dependent manner, which in principle compensates for the impaired electron entry of NADH into electron transfer chain (ETC) that utilizes oxygen as the terminal acceptor; in turn, PYCR1 activity is demonstrated to be essential for the tumor cells proliferation under hypoxia.

As a transmembrane tyrosine kinase, Insulin-like growth factor-1 receptor (IGF1R) vitally participates in the regulation of cell growth and differentiation, and its dysregulation contributes to oncogenic transformation and cancer development[5,6]. In addition to the distribution at the plasma membrane, IGF1R is able to translocate into the nucleus following SUMOylation and clathrin-mediated endocytosis[7]. It has been reported that IGF1R exhibits regulatory effects on multiple nuclear activities including DNA repair and replication and gene transcription, which is realized through the kinase activity of IGF1R or

¹Department of Radiotherapy, The Affiliated Cancer Hospital of Zhengzhou University & Henan Cancer Hospital, Zhengzhou, China. ²Department of Liver Surgery and Shanghai Cancer Institute, State Key Laboratory of Systems Medicine for Cancer, Renji Hospital, Shanghai Jiao Tong University School of Medicine, Shanghai, China. ³Endoscopy Center, Department of Gastroenterology, Shanghai East Hospital, School of Medicine, Tongji University, Shanghai, China. ⁴Department of Pathology, Shanghai East Hospital, School of Medicine, Tongji University, Shanghai, China. ⁵These authors contributed equally: Ke Zheng, Nannan Sha. ✉e-mail: 1800512@tongji.edu.cn; yhjiang@shsmu.edu.cn; chentao@tongji.edu.cn

its binding ability to protein components and DNA sequence at promoter region[8–12], and the relevant physiological influence of nuclear IGF1R correlates significantly with tumor malignancy[9,13,14]. IGF1R is highly expressed in tumor tissues in distinct cancers, in which hypoxia would be one of the important stimulus[15–17], and IGF1R is potentially responsible for tumor cell growth or metastasis under stress signals[15,17], although the underlying mechanism remains needs to be further explored.

In this study, we show that hypoxia promotes IGF1R-catalyzed PYCR1 phosphorylation in the nucleus, and this phosphorylation causes the complex formation between PYCR1 and ELK4 with its enrichment at ELK4 target genes promoter. Based on the binding to ELK4, the enzymatic activity of PYCR1 for NADH oxidation facilitates ELK4-Sirt7-mediated genes transcription repression and thus prevents cell growth arrest under hypoxia, in which NAD⁺ generation by PYCR1 locally potentiates deacetylase activity of Sirt7 that is critical for transcription inhibition. Further physiological analysis demonstrates the important relevance of the compartmental regulation by PYCR1 to CRC development.

## Results

### The nuclear localization of PYCR1 is critical for tumor cell growth under hypoxia

To probe the unknown effect of PYCR1 on tumor cell growth under hypoxia, the expression and subcellular distribution of PYCR1 were first examined across 150 human colorectal tumor specimens and multiple colorectal tumor cell lines. Immunohistochemistry (IHC) indicated the level of PYCR1 was largely elevated in tumor tissues compared with the adjacent normal tissues (Fig. 1a). Further IHC (Supplementary Fig. 1a) and cellular fractionation (Fig. 1b and Supplementary Fig. 1b) analysis indicated PYCR1 was constantly localized in the nucleus of tumor cells with a notable level regardless of oxygen condition. Prediction analysis using cNLS mapper[18] revealed amino sequence of ²⁸HKIMASSPDMDLATVSALRKMGVKLTPH⁵⁵ (N1) and ²⁸²IKKTILDKVKLDSPAGTALSPSGHTKLLP³¹⁰ (N2) were PYCR1 putative nuclear location signals, and mutagenesis analysis indicated mutation of arginine and lysine of N1 into alanine greatly diminished PYCR1 nuclear distribution (Fig. 1c and Supplementary Fig. 1c). These observations imply a potential function of nuclear PYCR1. Given this, the Flag-tagged PYCR1 was stably expressed in HCT116 cells, and immunoprecipitation of nuclear fraction combined with the following mass spectrometry analysis revealed hypoxia specifically promoted the binding of PYCR1 to ELK4 and IGF1R (Fig. 1d and Supplementary Fig. 1d). The following immunoprecipitation analysis verified the interaction at endogenous level and indicated the complex formation was specifically occurred in the nucleus (Fig. 1e and Supplementary Fig. 1e, f). Then PYCR1, ELK4 and IGF1R were silenced respectively by the validated shRNAs in HCT116 cells (Supplementary Fig. 1g). It was found that depletion of IGF1R and PYCR1 mutually attenuated their interaction with ELK4 (Fig. 1f), while ELK4 depletion had no effect on IGF1R-PYCR1 interaction (Supplementary Fig. 1h). Consistent with previous finding[19], the nuclear accumulation of IGF1R was largely enhanced in distinct tumor cell lines under hypoxia, which was accompanied by the increased level of total protein and phosphorylation that indicates IGF1R activation (Supplementary Fig. 1i). We doubted whether the kinase activity of IGF1R is involved in ELK4-PYCR1 interaction. The inhibitors against the kinases whose activity known to be responsive to hypoxia were collected, by which the immunoprecipitation analysis indicated PYCR1-ELK4 complex formation was blocked by PPP, an inhibitor against IGF1R, but not by ARQ-092 and GDC-0994 that were utilized as inhibitors against AKT and ERK (Fig. 1g and Supplementary Fig. 1j), suggesting IGF1R activity is importantly involved. Of note, cellular fractionation analysis indicated neither IGF1R nor ELK4 depletion could affect the nuclear level of PYCR1 (Supplementary Fig. 1k). Subsequently, the physiological effect of

nuclear PYCR1 was examined. The shRNA-resistant wild-type PYCR1 (WT rPYCR1) or PYCR1 mN1 (rPYCR1 mN1) were reconstitutively expressed in HCT116 and SW620 cells (Supplementary Fig. 1l), and the functional analysis indicated expression of rPYCR1 mN1, which exhibited comparable enzymatic activity to WT PYCR1 as compared with the enzymatic dead mutant PYCR1 T238A (Fig. 1h), dramatically reduced cell growth under hypoxia (Fig. 1i and Supplementary Fig. 1m). In contrast, PYCR1 depletion resulted in an impairment of cell growth under normoxia, although the extent of which was shown to be lesser than that under hypoxia (Supplementary Fig. 1n, o). Meanwhile, it was found that the negative effect of rPYCR1 mN1 (Fig. 1i and Supplementary Fig. 1m) or PYCR1 depletion (Supplementary Fig. 1n, o) on cell growth was unable to be restored by supplementation of exogenous proline. These results indicate nuclear location of PYCR1 is required for sustained cell growth under hypoxia, which is independent of its metabolic activity for proline generation.

### IGF1R phosphorylates PYCR1 and promotes PYCR1-ELK4 interaction

We set forth to investigate the mechanism underlying the regulation of PYCR1-ELK4 interaction by IGF1R. Immunoprecipitation analysis showed CIP treatment in precipitates from HCT116 and SW620 cells evidently disrupted PYCR1-ELK4 interaction under hypoxia (Fig. 2a and Supplementary Fig. 2a), revealing the importance of protein phosphorylation for the complex formation. Along with the effect of IGF1R activity on PYCR1-ELK4 interaction (Fig. 1g), it could be assumed that IGF1R might promote their binding by targeting PYCR1 or ELK4 as the kinase substrate. In support of this assumption, it was found the tyrosine residue of PYCR1 could be phosphorylated by IGF1R during the in vitro kinase assay (Fig. 2b, the 3rd and 5th lanes). The His-pull down analysis indicated both CIP-treated and -untreated cellular ELK4 could not bind to purified PYCR1 alone (Fig. 2b, the 2nd and 4th lanes), while successfully interacting with PYCR1 that underwent IGF1R kinase assay (Fig. 2b, the 3rd and 5th lanes). These data suggest the IGF1R can phosphorylate PYCR1, which facilitates its binding to ELK4. To identify the phosphor-site of PYCR1 by IGF1R, Tyr-135 and Tyr-180, the only tyrosine residues contained in the amino acids sequence of PYCR1, were mutated into phenylalanine respectively (PYCR1 Y135F and PYCR1 Y180F), and the following kinase assay indicated mutation of PYCR1 Tyr-135, a protein surface-locating residue (Supplementary Fig. 2b), instead of PYCR1 Tyr-180, was resistant to IGF1R-mediated phosphorylation (Fig. 2c), indicating PYCR1 Tyrosine 135 is the major IGF1R-phosphor-site. Meanwhile, PYCR1 Tyr-135 phosphorylation (PYCR1 pY135) was demonstrated by an antibody that can specifically recognize it (Fig. 2c). Immunoblotting analysis indicated PYCR1 pY135 was evidently induced and primarily detected in the nucleus under hypoxia in HCT116 (Fig. 2d) and SW620 (Supplementary Fig. 2c, left panel) cells, which was diminished by either PPP treatment (Fig. 2d and Supplementary Fig. 2c, left panel) or IGF1R depletion (Fig. 2e and Supplementary Fig. 2c, right panel). In line with this, the nuclear accumulation of PYCR1 pY135 was largely compromised in rPYCR1 mN1-expressing cells (Supplementary Fig. 2d, e), and the inductive effect of IGF1R on PYCR1 pY135 under hypoxia was also validated by the immunofluorescence analysis (Supplementary Fig. 2f). Further analysis showed that the amount of Tyr-135-phosphorylated PYCR1 accounted for around 35% of total level of nuclear PYCR1 under hypoxia (Supplementary Fig. 2g). Next, immunoprecipitation analysis indicated compared with WT PYCR1, PYCR1 Y135F largely lost the interaction with ELK4 under hypoxia (Fig. 2f and Supplementary Fig. 2h), although it displayed a comparable nuclear level to that of WT PYCR1 (Supplementary Fig. 2i). Then HCT116 (Fig. 2g) and SW620 (Supplementary Fig. 2j) cells were depleted of endogenous PYCR1 and reconstituted with shRNA resistant PYCR1 Y135F and PYCR1 T238A (rPYCR1 Y135F and rPYCR1 T238A), and the functional analysis showed expression of rPYCR1 Y135F (Fig. 2h and Supplementary Fig. 2k) and rPYCR1 T238A

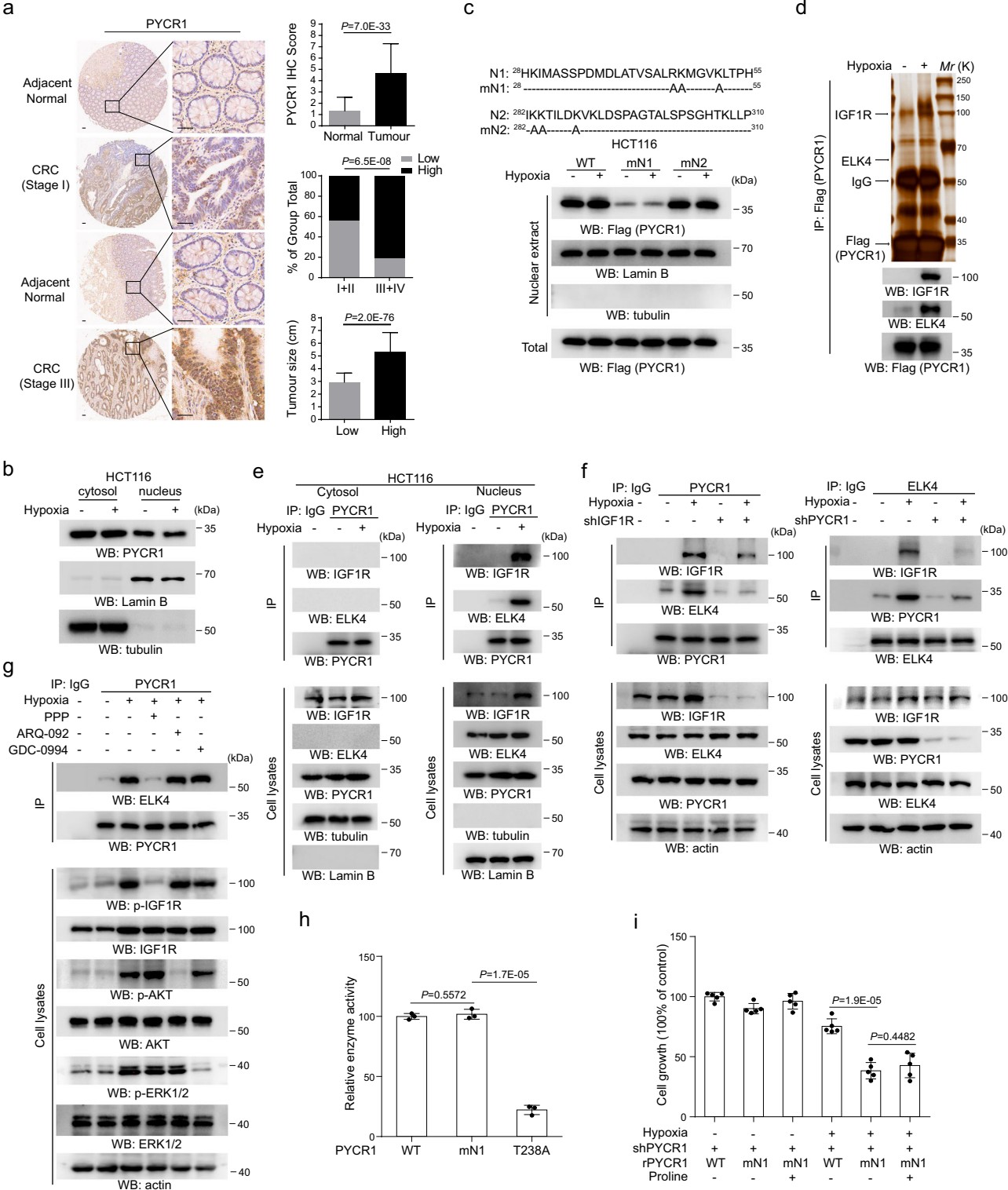

(Fig. 2i and Supplementary Fig. 2l) both dramatically decreased cell growth under hypoxia, in which ELK4 depletion (Supplementary Fig. 2m) suppressed cell growth with an overwhelming effect. Moreover, it was shown that PYCR1 Y135F but not PYCR1 T238A retained a similar enzymatic activity to WT PYCR1 (Supplementary Fig. 2n), which was also reflected by overall intracellular (Fig. 2j) or nuclear (Supplementary Fig. 2o) proline levels in each group. These results suggest either PYCR1 phosphorylation required for PYCR1-ELK4 interaction or enzymatic activity is positively involved in cell growth maintenance under hypoxia.

## PYCR1 facilitates ELK4-mediated transcriptional repression for growth maintenance

It has been known that ELK4 can act as a transcriptional repressor to support cell growth[20]. To determine whether PYCR1 can regulate ELK4-mediated transcription under hypoxia in colorectal tumor cells, the RNA-seq analysis was performed in WT rPYCR1- and rPYCR1 Y135F-expressing HCT116 cells under hypoxia, which indicated 656 genes were upregulated and 373 genes were downregulated by rPYCR1 Y135F expression (Supplementary Data file 1). The subsequent transcriptional motif analysis of promoter region (2 kb upstream of transcription start

**Fig. 1 | The nuclear localization of PYCR1 is critical for tumor cell growth under hypoxia. b–g** immunoblotting and immunoprecipitation analysis were performed using the indicated antibodies. **a, h, i** the values are presented as mean ± s.d.; statistical analysis was performed using the two-tailed Student's *t* test or Pearson's chi-squared test. Source data are provided as Source Data uncropped western blots and Source Data Fig. 1. **a** IHC analysis was performed in 150 human colorectal tumor specimens, representative images are shown (upper panel, *n* = 150; middle panel, for I+II, *n* = 81, in which low and high, respectively, are 45 and 36, for III+IV, *n* = 69, in which low and high, respectively, are 13 and 56; bottom panel, for Low, *n* = 58, for High, *n* = 92). Scale bars, 100 µm. **b** HCT116 cells were cultured under normoxia or hypoxia for 12 h and cytosolic and nuclear extracts were collected. **c** HCT116 cells expressing indicated Flag-tagged PYCR1 were cultured under hypoxia for 12 h, and nuclear extracts were collected. **d** HCT116 cells expressing Flag-tagged PYCR1 were

cultured under normoxia or hypoxia for 12 h. Nuclear extracts subjected to immunoprecipitation with the anti-Flag antibody were analyzed by silver staining. **e** HCT116 cells were cultured under normoxia or hypoxia for 12 h. Immunoprecipitation analysis was performed post collection of cytosolic (left panel) or nuclear extracts (right panel). **f** HCT116 cells transfected with IGF1R shRNA (left panel) or PYCR1 shRNA (right panel) were cultured under normoxia or hypoxia for 12 h. **g** HCT116 cells were pretreated with PPP (1 µM), ARQ-092 (2 µM) and GDC-0994 (10 µM) for 1 h before being cultured under hypoxia for 12 h. **h** WT and mutant His-PYCR1 were purified and the enzymatic activity was measured (*n* = 3 independent experiments). **i** HCT116 cells with depletion of PYCR1 and reconstituted with an expression of indicated Flag-rPYCR1 were treated with or without exogenous proline (3 mM). Cells were cultured under normoxia or hypoxia for 48 h. Cellular viability was examined by CCK-8 assay (*n* = 5 independent experiments).

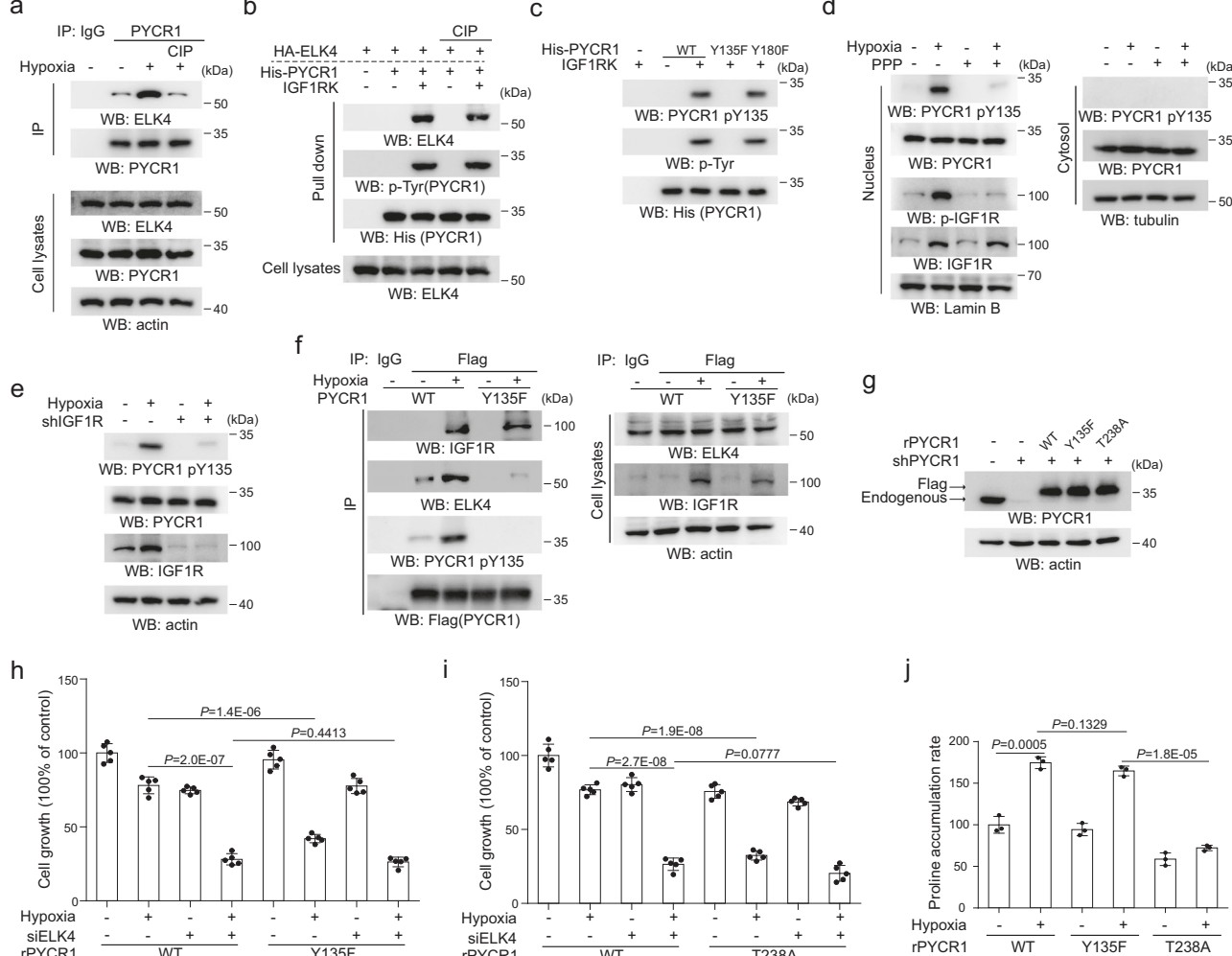

**Fig. 2 | IGF1R phosphorylates PYCR1 and promotes PYCR1-ELK4 interaction. a–f** immunoprecipitation and immunoblotting analysis were performed using the indicated antibodies. In **h–j**, the values are presented as mean ± s.d., statistical analysis was performed using the two-tailed Student's *t* test. Source data are provided as Source Data uncropped western blots and Source Data Fig. 2. **a** HCT116 cells were cultured under hypoxia for 12 h. The immunoprecipitates were treated with CIP (10 units) and analyzed by immunoblotting. **b** HA-ELK4 immunoprecipitated from HCT116 cells post hypoxia culturing was treated with or without CIP (10 units). Then immunoprecipitate was incubated with His-PYCR1 which was phosphorylated by IGF1RK and the His-pull down analysis was performed. **c** In vitro phosphorylation analysis were performed by mixing the purified IGF1RK with indicated purified His-PYCR1s proteins. **d, e** HCT116 cells were pretreated with or

without PPP (1 µM) for 1 h (**d**) or transfected with or without IGF1R siRNA (**e**). Cells were cultured under normoxia or hypoxia for 12 h and cytosolic or nuclear extracts were collected. **f** HCT116 cells expressing indicated Flag-PYCR1s were cultured under normoxia or hypoxia for 12 h. Immunoprecipitation analysis were performed. **g** HCT116 cells with depletion of endogenous PYCR1 were reconstitutively expressed with indicated Flag-rPYCR1s. **h, i** HCT116 cells with reconstituted expression of WT rPYCR1 and rPYCR1 Y135F (**h**) or rPYCR1 T238A (**i**) were transfected with or without ELK4 siRNA. Cells were cultured under normoxia or hypoxia for 48 h. Cellular viability was examined by CCK-8 assay (*n* = 5 independent experiments). **j** HCT116 cells with reconstituted expression of indicated rPYCR1 were cultured under normoxia or hypoxia for 24 h. The proline level was measured (*n* = 3 independent experiments).

site) revealed ELK4-binding consensus sequence (CCGGA or C/TTTCC) (Supplementary Fig. 3a) was presented in promoters of 242 upregulated genes and 139 downregulated genes (Supplementary Data file 2) under rPYCR1 Y135F expression (Fig. 3a), suggesting the binding of PYCR1 to ELK4 potentially affects the transcription of ELK4-target genes. We assumed the transcriptional effect of PYCR1 and ELK4 would be concordant with their functional relevance. Based on the inhibitory role in cell growth, four ELK4-targeted and growth-relevant genes, *KLK10, DEPP1, PTPRR,* and *CNN1*, shown to be upregulated in rPYCR1 Y135F group were selected in the following analysis. The transcriptional effect of ELK4 on the target genes was validated by the luciferase analysis (Supplementary Fig. 3b). As shown in Fig. 3b, the effect of rPYCR1 Y135F expression on genes transcription was verified through real-time PCR analysis, in which ELK4 depletion also resulted in a remarkable increase of transcription regardless of PYCR1 status under hypoxia. Meanwhile, ELK4-target genes transcription was enhanced by expression of rPYCR1 mN1 (Supplementary Fig. 3c) or blockade of IGF1R activity (Supplementary Fig. 3d). In line with the requirement of enzymatic activity for the cell growth, expression of rPYCR1 T238A that retained binding ability of ELK4 (Supplementary Fig. 3e), similarly increased transcription of ELK4-target genes under hypoxia (Fig. 3c). Furthermore, ChIP-qPCR analysis indicated the enrichment of ELK4 at genes promoter was significantly promoted under hypoxia (Fig. 3d), which remained unchanged under expression of rPYCR1 T238A or rPYCR1 Y135F. At the same condition, WT rPYCR1 and rPYCR1 T238A rather than rPYCR1 Y135F was notably accumulated at the selected genes promoter in an ELK4 dependent manner (Fig. 3e, f), which did not occur at promoter of *GAPDH* that is recognized as a negative control (Supplementary Fig. 3f). Above all, these results suggest both ELK4-binding ability and metabolic activity of PYCR1 are required for transcription repression of ELK4-target genes under hypoxia.

## PYCR1 facilitates Sirt7-mediated H3K18 deacetylation

Sirtuin 7 (SIRT7) has been implicated in ELK4-mediated suppression of genes transcription[21]. Immunoprecipitation analysis indicated Sirt7 basally interacted with ELK4 at a notable level, which was mildly augmented under hypoxia to a similar level in WT rPYCR1- and rPYCR1 Y135F-expressing cells (Fig. 4a and Supplementary Fig. 4a). Meanwhile, the binding of PYCR1 to SIRT7 was found to be increased under hypoxia in an ELK4 dependent manner (Supplementary Fig. 4b). Evidently, depletion of Sirt7 (Supplementary Fig. 4c) largely increased expression of *KLK10, DEPP1, PTPRR,* and *CNN1* regardless of PYCR1 status at condition of hypoxia (Fig. 4b, c), in which no synergic effect was observed upon Sirt7 depletion and expression of rPYCR1 Y135F or rPYCR1 T238A, revealing the essential role of Sirt7 in the regulation of ELK4-target genes transcription. Accordingly, ChIP-qPCR analysis showed the enrichment of Sirt7 at ELK4-target genes promoter was significantly increased under hypoxia and was showed to be relied on ELK4, and these effects were unaffected by expression of rPYCR1 mutants (Fig. 4d).

SIRT7 displays deacetylase activity against H3K18Ac (H3K18 acetylation) that marks transcriptional activation, thus we speculated this would be relevant to the transcriptional effect of Sirt7 on ELK4-targeted genes. In support of this assumption, overexpression of H3K18R (Fig. 4e) blocked gene expression either in Sirt7-depleted (Fig. 4f) or ELK4-depleted (Supplementary Fig. 4d) cells under hypoxia. Likewise, the transcriptional induction by rPYCR1 Y135F or rPYCR1 T238A expression was abrogated by H3K18R overexpression under hypoxia (Fig. 4g). The limited transcriptional effects of H3K18R overexpression on these genes under normoxia (Supplementary Fig. 4e, f) would be attributed to their weak basal expression or H3K18Ac levels. ChIP-qPCR analysis indicated the promoter-associated enrichment of H3K18Ac under hypoxia was found to be enhanced upon depletion of Sirt7 (Fig. 4h) or ELK4 (Supplementary Fig. 4g). Meanwhile, H3K18Ac enrichment at promoter of ELK4-target genes (Fig. 4i) instead of *GAPDH* (Supplementary Fig. 4h) was augmented as well in rPYCR1

Y135F- or rPYCR1 T238A-expressing cells. Moreover, it was found that the promoter-associated level of H3K36Ac (H3K36 acetylation), an alternative substrate of SIRT7 known to be related to active gene promoters[22], was also increased by Sirt7 depletion (Supplementary Fig. 4i) or rPYCR1 mutants expression (Supplementary Fig. 4j) for ELK4-targeted gene *KLK10*. These results suggest IGF1R-ELK4-PYCR1 axis promotes Sirt7-mediated histone deacetylation, and thus exert a repressive effect on genes transcription.

## PYCR1-catalyzed NAD$^+$ production is involved in transcriptional regulation

The final step of proline synthesis conducted by PYCR1 is associated with NADH oxidation and thus NAD$^+$ production. Further analysis indicated neither transcription (Fig. 5a) nor promoter-associated H3K18Ac enrichment of ELK4-targeted genes (Fig. 5b) were changed by addition of exogenous proline in rPYCR1 Y135F- or rPYCR1 T238A-expressing cells under hypoxia. In contrast with that of exogenous proline, overexpression of NMNAT-1 (Supplementary Fig. 5a), which substantially increased nuclear level of NAD$^+$ (Supplementary Fig. 5b) under hypoxia, partially reversed genes transcription and promoter-associated H3K18Ac accumulation in rPYCR1 Y135F- or rPYCR1 T238A-expressing cells (Fig. 5a, b) as well as in PYCR1-depleted cells (Supplementary Fig. 5c, d), suggesting NAD$^+$ instead of proline availability is involved in genes transcription regulation by PYCR1. It could be assumed that mutation of PYCR1 Y135F uncouples the proximal metabolic activity of PYCR1 for ELK4-mediated transcriptional activity and thereby results in the change of gene expression, while NMNAT-1 overexpression produces a compensatory effect in this regard. To further validate the relevance of PYCR1 metabolic activity to genes transcription, P5CS, the upstream enzyme of PYCR1 responsible for the conversion of glutamate to $\Delta^1$-pyrroline-5-carboxylic acid (P5C) (Fig. 5c), was depleted in HCT116 cells (Supplementary Fig. 5e). As a result, P5CS depletion, which largely decreased the nuclear amount of proline (Supplementary Fig. 5f), notably increased transcription of ELK4-targeted genes (Fig. 5d) as well as promoter enrichment of H3K18Ac (Fig. 5e), and significantly impaired of cell growth under hypoxia (Supplementary Fig. 5g). Of note, further analysis indicated P5CS was basally localized in the nucleus (Supplementary Fig. 5h), and its nuclear level was moderately increased by hypoxia treatment (Supplementary Fig. 5h), which was accompanied by an enhancement of the enzymatic activity (Supplementary Fig. 5i). These results reveal the upstream activity of proline synthesis pathway is required for PYCR1-related transcriptional regulation under hypoxia. We wondered if PYCR1-catalyzed NAD$^+$ production fuels Sirt7-mediated histone deacetylation at genes promoter. To clarify this issue, chromatins were collected from P5CS-depleted, PYCR1-depleted, or rPYCR1 T238A-expressing HCT116 cells after culturing under hypoxia, which was incubated with according metabolites in a buffer suite for in vitro PYCR1-enzymatic reaction and the subsequent Sirt7-mediated deacetylation assay. As a result, promoter-associated accumulation of H3K18Ac induced upon P5CS-depleted cells was successfully deacetylated by addition of NAD$^+$ alone or P5C combined with NADH in a Sirt7-dependent manner (Fig. 5f). In contrast, only the deacetylation effect from NAD$^+$ rather than P5C-NADH incubation could be detected in PYCR1-depleted cells (Supplementary Fig. 5j), suggesting the metabolic activity of PYCR1 for NAD$^+$ production is critical for Sirt7-mediated H3K18 deacetylation. Consistently, for chromatins collected from rPYCR1 T238A-expressing cells, the intensity of H3K18Ac at the promoter region was readily removed by NAD$^+$ incubation in a Sirt7-dependent manner (Fig. 5g).

## PYCR1-Y135 phosphorylation facilitates tumor development in colorectal cancer

To determine the potential implication of PYCR1 pY135 in tumor growth in vivo, HCT116 or SW620 cells with or without PYCR1

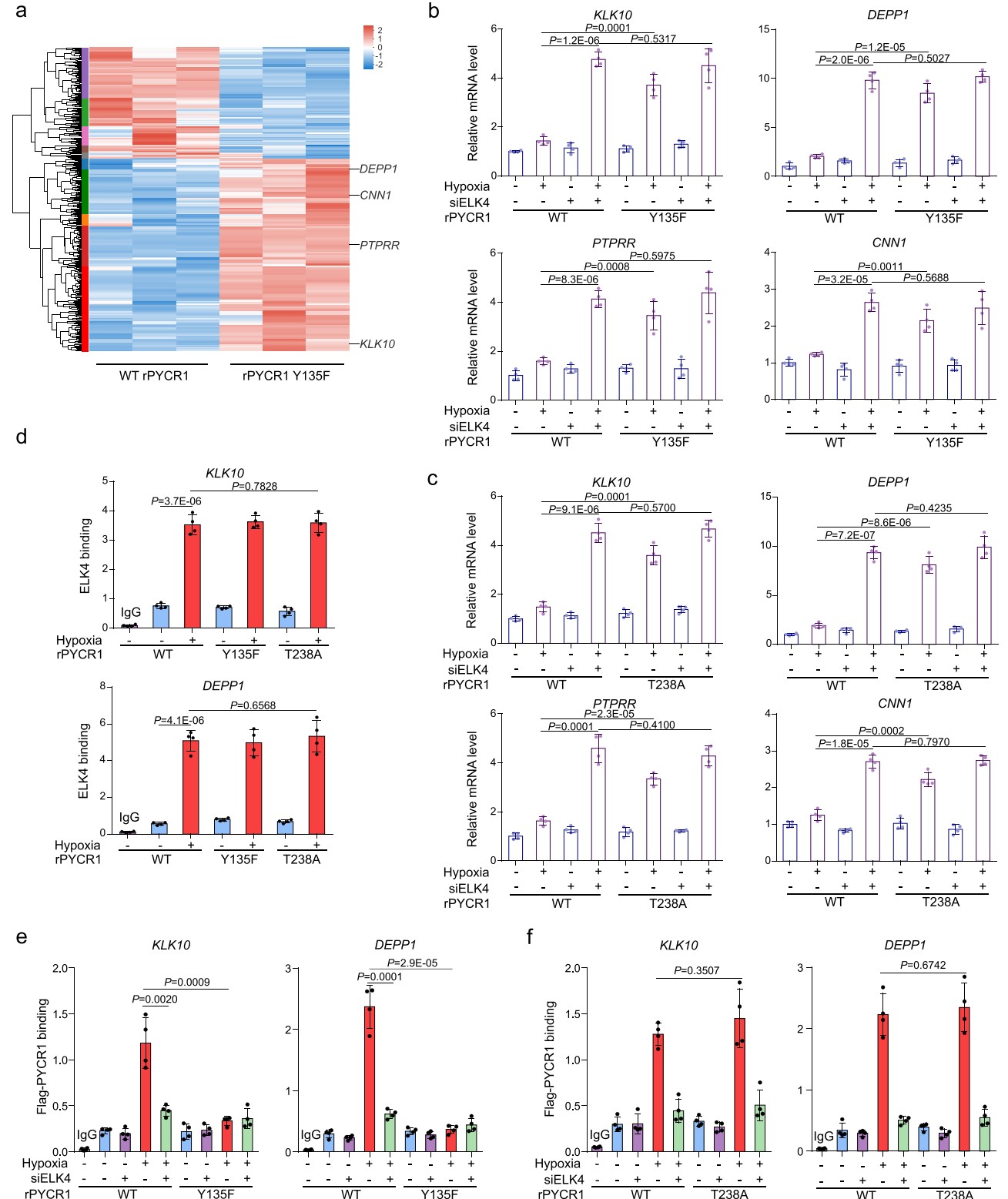

**Fig. 3 | PYCR1 facilitates ELK4-mediated transcriptional repression for growth maintenance. a–f** HCT116 cells expressing the indicated Flag-rPYCR1 were collected. **b**, **c**, **e**, **f** cells were simultaneously transfected with or without ELK4 siRNA. **b**, **c** mRNA levels of indicated ELK4 target genes were analyzed by real-time PCR. **d–f** ChIP analysis were performed using the indicated antibody; the primers covering the ELK4-binding sites at promoter regions of the indicated genes were utilized for real-time PCR analysis. The y axis shows the value normalized to the input. **a** differential expression analysis was performed using the edgeR. **b–f** data are presented as the mean ± s.d. (n = 4 independent experiments); statistical analysis

was performed using the two-tailed Student's t test. Source data are provided as Source Data Fig. 3. **a** The cDNA microarray analysis was performed for cells cultured under hypoxia for 24 h. Among genes that were expressed differentially between two groups, the genes whose promoter contained the ELK4-binding consensus were presented. **b**, **c** Cells expressing WT rPYCR1 and rPYCR1 Y135F (**b**) or T238A (**c**) were cultured under hypoxia for 24 h. **d** Cells expressing the indicated Flag-rPYCR1 were cultured under hypoxia for 12 h. **e**, **f** Cells expressing WT rPYCR1 and rPYCR1 Y135F (**e**) or T238A (**f**) were cultured under hypoxia for 12 h.

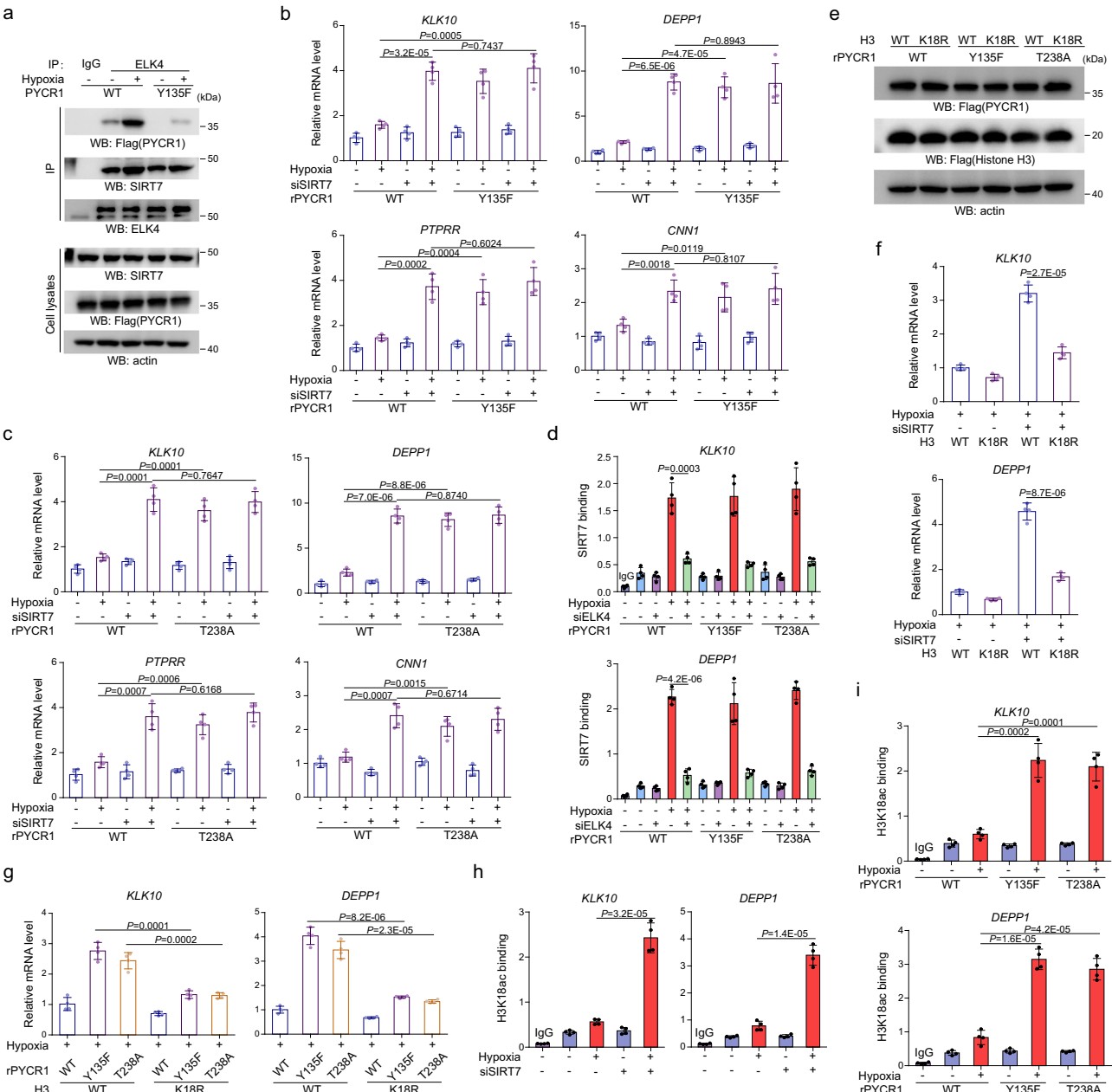

**Fig. 4 | PYCR1 facilitates Sirt7-mediated H3K18 deacetylation.**
**a**, **e** immunoprecipitation and immunoblotting analysis were performed with indicated antibodies. **d**, **h**, **i** ChIP analysis were performed with indicated antibodies, the primers covering the ELK4-binding sites at promoter regions of the indicated genes were utilized for real-time PCR analysis. The *y* axis shows the value normalized to the input. **b**, **c**, **f**, **g** mRNA levels of indicated ELK4 target genes were analyzed by real-time PCR. **b–d**, **f–i** data are presented as the mean ± s.d. (*n* = 4 independent experiments); statistical analysis was performed using the two-tailed Student's *t* test. Source data are provided as Source Data uncropped western blots and Source Data Fig. 4. **a** HCT116 cells expressing indicated Flag-PYCR1s were cultured under normoxia or hypoxia for 12 h. **b**, **c** HCT116 cells expressing WT

rPYCR1 and rPYCR1 Y135F (**b**) or T238A (**c**) were transfected with or without Sirt7 siRNA. Cells were cultured under normoxia or hypoxia for 24 h. **d** HCT116 cells expressing the indicated Flag-rPYCR1s were transfected with or without ELK4 siRNA. Cells were cultured under normoxia or hypoxia for 12 h. **e** HCT116 cells expressing the indicated Flag-rPYCR1s were overexpressed with WT Histone H3 or Histone H3K18R. **f**, **g** HCT116 cells transfected with or without Sirt7 siRNA (**f**) or expressing indicated Flag-rPYCR1s (**g**) were overexpressed with WT Histone H3 or Histone H3K18R. Cells were cultured under normoxia or hypoxia for 24 h. **h**, **i** HCT116 cells transfected with or without Sirt7 siRNA (**h**) or expressing indicated Flag-rPYCR1s (**i**) were cultured under normoxia or hypoxia for 12 h.

depletion, or with respective expression of WT rPYCR1, rPYCR1 Y135F or rPYCR1 T238A were subcutaneously injected into athymic nude mice. In agreement with the previous findings[2], PYCR1 depletion resulted in a large impairment of tumor growth (Supplementary Fig. 6a, b). Compared with that of WT rPYCR1, tumor growth was also notably attenuated by expression of rPYCR1 Y135F or rPYCR1 T238A (Fig. 6a and Supplementary Fig. 6c), revealing the importance of Tyr-135 phosphorylation and enzymatic activity of PYCR1. Consistent

with the observations in cell culture under hypoxia, immunoblotting analysis of tumor tissues indicated expression of both rPYCR1 Y135F and rPYCR1 T238A notably increased the protein level of ELK4-target genes like *KLK10* and *CNN1* (Fig. 6b and Supplementary Fig. 6d), in which the elevated IGF1R phosphorylation was associated with the induction of PYCR1-Y135 phosphorylation in groups of WT rPYCR1 and rPYCR1 T238A. Meanwhile, the level of PYCR1 pY135 was shown to correlate with HIF-1α levels in mice tumor tissues (Supplementary

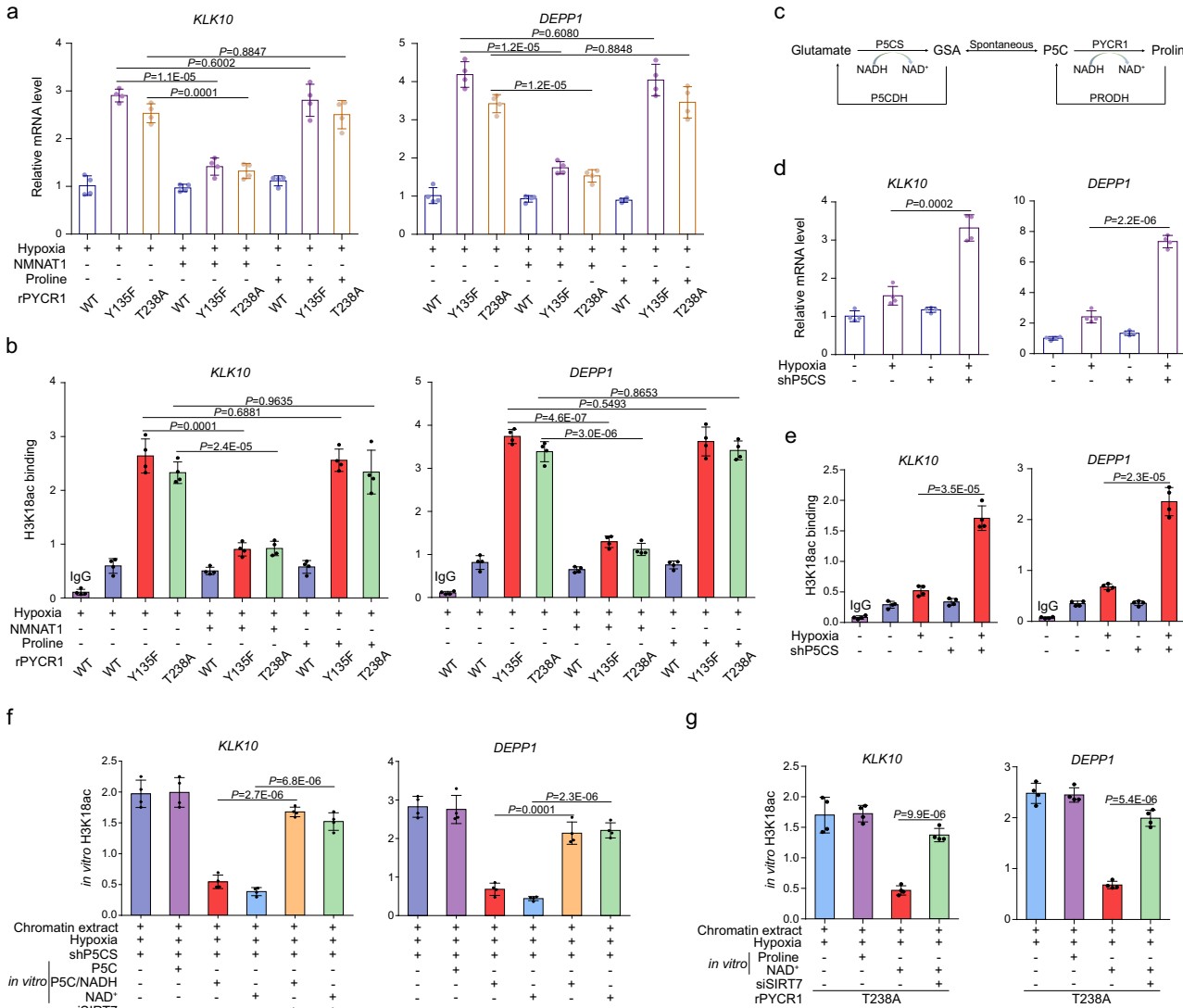

**Fig. 5 | PYCR1-catalyzed NAD⁺ production is involved in transcriptional regulation. a, d** mRNA levels of indicated ELK4 target genes were analyzed by real-time PCR. In b and e-g, the primers covering the ELK4-binding sites at promoter regions of the indicated genes were utilized for real-time PCR analysis. The $y$ axis shows the value normalized to the input. **a, b, d–g** data are presented as the mean ± s.d. ($n = 4$ independent experiments); statistical analysis was performed using the two-tailed Student's $t$ test. Source data are provided as Source Data Fig. 5. **a, b** HCT116 cells expressing the indicated Flag-rPYCR1s were simultaneously overexpressed with or without NMNAT-1. Cells were supplemented with the metabolites as indicated.

Cells were cultured under hypoxia for 24 h (**a**) or 12 h (**b**). Real-time PCR (**a**) or ChIP (**b**) analysis was performed. **c** The cartoon showing the key enzymes involved in proline synthesis from glutamine. **d, e** HCT116 cells with or without P5CS depletion were cultured under normoxia or hypoxia for 24 h (**d**) or 12 h (**e**). Real-time PCR (**d**) or ChIP (**e**) analysis was performed. **f, g** HCT116 cells with depleted P5CS (**f**) or with expression of rPYCR1 T238A (**g**) were transfected with or without Sirt7 siRNA. Cells were cultured under hypoxia for 12 h. Chromatin extracts were collected and mixed with or without the indicated metabolites for 30 min, and then chromatin extracts were fixed and used for ChIP analysis.

Fig. 6e). Then, the clinical relevance of PYCR1 pY135 was further evaluated in 150 colorectal cancer patient samples. The IHC analysis was performed following the validation of PYCR1-Y135 phosphor-antibody using the specific blocking peptides (Supplementary Fig. 6f). It was shown that either PYCR1 pY135 or IGF1R phosphorylation levels were increased in tumor tissues compared with the adjacent normal tissues (Fig. 6c, d). Moreover, the level of PYCR1 pY135 and IGF1R phosphorylation were found to be significantly correlated with the size and malignancy of tumors (Fig. 6e, f). As expected, the level of PYCR1 pY135 and IGF1R phosphorylation were significantly correlated (Supplementary Fig. 6g); the signals of PYCR1 pY135 also exhibited a positive relationship with that of HIF-1α in distinct tumor regions (Supplementary Fig. 6h). These data suggest IGF1R-mediated PYCR1 pY135 exerts a tumor growth-promoting effect on colorectal cancer development.

## Discussion

PYCR1 is responsible for the last step of proline synthesis through catalyzing NAD(P)H-dependent reduction of pyrroline-5-carboxylate. Although PYCR1 is conventionally considered to be mainly distributed in the mitochondrial matrix, either immunoblotting or IHC analysis in our study indicates PYCR1 constitutively localizes in the nucleus of tumor cells, implying that PYCR1 would exert a potential subcellular effect on nuclear activities. Given this, we further found hypoxia induces phosphorylation of PYCR1 by IGF1R in the nucleus, which led to PYCR1-ELK4 interaction and their accumulation at ELK4-target genes promoter. Within the protein complex, PYCR1 catalyzed NADH oxidation promotes Sirt7 deacetylase activity and thereby facilitates genes transcription repression that maintains cell growth under hypoxia (Fig. 6g). These findings illustrate an unidentified regulatory mechanism of gene transcription by nuclear PYCR1 and point to PYCR1

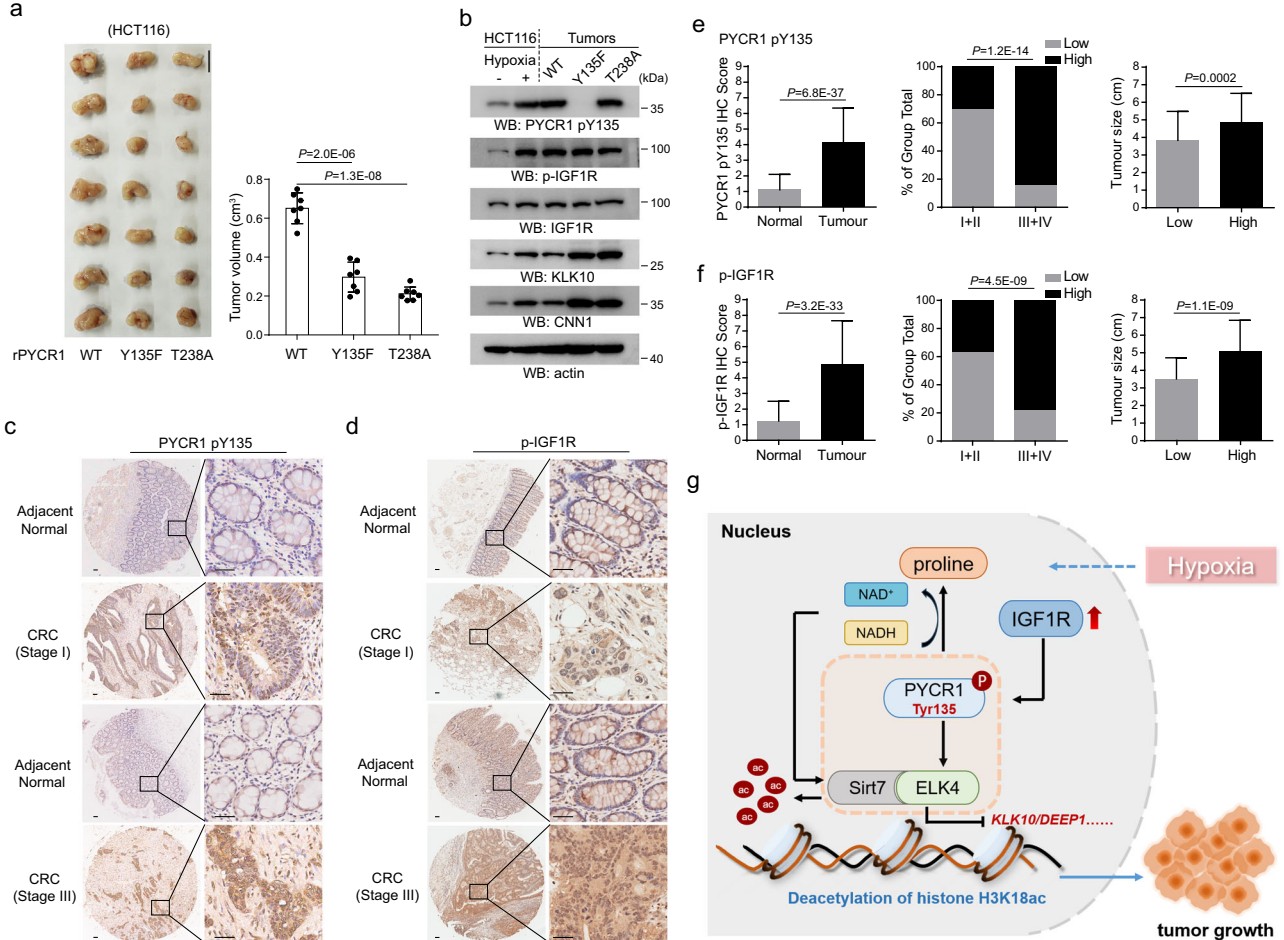

**Fig. 6 | PYCR1-Y135 phosphorylation facilitates tumor development in colorectal cancer. a, e, f** the values are presented as mean ± s.d.; statistical analysis was performed using the two-tailed Student's *t* test or Pearson's chi-squared test. Source data are provided as Source Data uncropped western blots and Source Data Fig. 6. **a** A total of $3 \times 10^6$ HCT116 cells expressing the indicated Flag-rPYCR1 were subcutaneously injected into the athymic nude mice. Representative tumor xenografts were shown (left panel). Scale bars: 1 cm. Tumor volumes were measured by using length "a" and width "b" and calculated using the following equation: $V = ab^2/2$. Data are represented on right panel ($n = 7$). **b** Lysates collected from HCT116 cells cultured under hypoxia and from tumor tissues were subjected to immunoblotting analysis. **c, d** IHC staining with anti-PYCR1 pY135 (**c**) or anti-p-IGF1R (**d**) antibody was performed on 150 human colorectal cancer specimens. Representative photos of early-stage and late stage tumor versus the adjacent normal tissues were shown (magnification: ×40 and ×200). Scale bars: 100 μm. **e, f** The

*t* test indicating a significant difference in PYCR1 pY135 (**e**) or p-IGF1R (**f**) level between tumor and the adjacent normal tissues (left panels, $n = 150$ human colorectal tumor specimens). The chi-square test indicates significant association of high PYCR1 pY135 (**e**, middle panel, for I+II, $n = 81$, in which low and high respectively are 57 and 24, for III+IV, $n = 69$, in which low and high, respectively, are 11 and 58) or p-IGF1R (**f**, middle panel, for I+II, $n = 81$, in which low and high, respectively, are 51 and 30, for III+IV, $n = 69$, in which low and high, respectively, are 15 and 54) level with the number of early and late tumors. The *t* test indicated significant difference in tumor size between low level-groups and high level-groups of PYCR1 pY135 (**e**, right panel, for Low, $n = 58$, for High, $n = 92$) or p-IGF1R (**f**, right panel, for low, $n = 66$, for High, $n = 84$). **g** Schematic of the regulatory effect of IGF1R-phosphorylated PYCR1 on ELK4 transcriptional activity and its influence on tumor growth.

as the potential therapeutic target during tumor treatment at the physiological condition.

Consistent with previous studies[23], the overall expression level of IGF1R is found to be upregulated under hypoxia, and this response is supposed to be beneficial for tumor cell growth. Concomitantly, an increased level of nuclear IGF1R is detected under hypoxia, while its potential metabolic effect is rarely investigated before. Here PYCR1 is demonstrated to be the direct phosphor-target of IGF1R and acts as an important transducer of the stress-responsive signal initiated by IGF1R, which thus connects the physiological effect of IGF1R with cellular metabolic status.

The enhanced proline synthesis has been known to be vital for cell proliferation under hypoxia, in which PYCR1-catalyzed NADH oxidation is shown to be critically involved as it metabolically sustains TCA cycle when ETC function is hampered[3]. Along with our observations, it can be concluded that the enzymatic activity of PYCR1 for NADH

oxidation in the mitochondria and nucleus are both essential for accomplishing the supportive role of proline synthesis in cell growth under hypoxia, in which the downstream physiological influences are linked with metabolic and nonmetabolic activities respectively. In line with the significance of proline synthesis pathway, it is shown that P5C generation mediated by P5CS, which fuels PYCR1-catalyzed reaction, is accordingly indispensable for gene transcription regulation by PYCR1. However, whether the metabolic activity of P5CS can be subcellularly allocated or regulated by the stress signals remains unclear, and further investigation in this regard will be meaningful for a better understanding of how proline synthesis pathway is coordinated in a context-dependent manner. In addition, although we found the nuclear localization of PYCR1 relies on its putative NLS-sequence, the regulatory mechanism of PYCR1 nuclear translocation needs to be further elucidated. The notable basal level of nuclear PYCR1 and its insensitivity to hypoxia reveal there are unknown regulatory factors

relevant to PYCR1 subcellular distribution; this also implies PYCR1 would be able to associate with other unknown components to regulate according to activities in the nucleus.

## Methods

### Animal study

Approval for animal experiments was obtained from the Animal Care and Use Committee of Tongji University under protocol number TJBB05122101. Tumor size did not exceed 20 mm in any direction and tumor volume did not exceed 2000 mm$^3$. Mice were maintained in a temperature-controlled and light-controlled environment with ad libitum access to water. The mice were randomly put into separate groups/cages for experiments and received a standard chow diet (laboratory diet 5L0D; 58% of calories from carbohydrate, 13.5% of calories from fat, and 28.5% of calories from proteins). Five-week-old male nu/nu mice (seven per group) were subcutaneously injected with $3 \times 10^6$ gene-modified HCT116 or $5 \times 10^6$ gene-modified SW620 cells in 200 μl of PBS. Tumor volume was measured by using length (a) and width (b) with Vernier calipers and calculated using the equation: $V = ab^2/2$.

### Cell culture

HCT116 cells were maintained in McCoy's 5a Medium Modified supplemented with 10% fetal bovine serum (FBS). SW620 and SW480 cells were maintained in Leibovitz's L-15 Medium supplemented with 10% FBS. Hypoxia treatment was performed under 1% $O_2$ and 5% $CO2$ (Ruskinn Bugbox Plus). All cell lines were obtained from the American Type Culture Collection and authenticated by fingerprinting of short tandem repeats. All cell lines were routinely tested to ensure they were negative for mycoplasma contamination using PCR.

### Antibodies

Antibodies that recognize PYCR1 (13108-1-AP; 1:2000 for WB, 4 μg for IP, 1:100 for IF), P5CS (17719-1-AP; 1:1000 for WB), HIF-1α (20960-1-AP; 1:1000 for WB) and Tubulin (11224-1-AP; 1:5000 for WB) were purchased from Proteintech. Antibodies that recognize ELK4 (sc-166823; 1:800 for WB, 6 μg for IP, 15 μg for ChIP), IGF1R (sc-81464; 1:500 for WB), p-IGF1R (sc-135767; 1:500 for WB), SIRT7 (sc-365344; 1:800 for WB, 20 μg for ChIP), KLK10 (sc-100551; 1:500 for WB), CNN1 (sc-58707; 1:1000 for WB), BIP (sc-13539; 1:1000 for WB) and Lamin B (sc-365962; 1:1000 for WB) were purchased from Santa Cruz Biotechnology. Antibodies that recognize His (T505; 1:1000 for WB), AKT (phospho-ser473) (11054; 1:1000 for WB), ERK1/2 (29162; 1:1000 for WB) and ERK1/2 (phospho-Thr202/Tyr204) (12082; 1:1000 for WB) were purchased from Signalway Antibody. Antibody that recognizes AKT (9272 S; 1:1000 for WB) was purchased from Cell Signaling Technology. Antibodies that recognize PYCR1 (ab103314; 1:200 for IHC), Phospho-tyrosine (ab179530; 1:1000 for WB), HIF-1α (ab243860; 1:100 for IF) and H3K18ac (ab40888; 10 μg for ChIP) were purchased from Abcam. Antibody that recognizes β-actin (AC026; 1:1000 for WB) was purchased from ABclonal. Antibody that recognizes Flag (F1804; 1:1000 for WB, 4 μg for IP, 10 μg for ChIP) was purchased from Sigma-Aldrich. Rabbit polyclonal PYCR1 pTyr135 (1:800 for WB, 1:100 for IHC, 1:50 for IF) antibody was made by Signalway Antibody.

### Materials

PPP (HY-15494), GDC-0994 (HY-15947), and ARQ-092 (HY-19719) were purchased from MedChemExpress. Proline (P0380), NAD$^+$ (N3014), and NADH (N8129) were purchased from Sigma-Aldrich.

### DNA construction and RNA interfering

The DNA constructs were performed as previously described[24]. The DNA sequence encoding PYCR1 was inserted into the pLenti-puro or pLVX-neo vector and a 3 × Flag-tag was fused to the C terminus. The shRNAs were generated by cloning the target sequence into the pGIPZ

vector: human *PYCR1* shRNA_1 (sense, 5′-TGCACGGAGGTGGAAGAG G-3′) and shRNA_2 (sense, 5′-CTTCATCCTGGATGAAATA-3′); human *ELK4* shRNA_1 (sense, 5′-CAGACCTACAGAAGACATA-3′) and shRNA_2 (sense, 5′-AGGACTCAAGTTGGACTAT-3′); human *IGF1R* shRNA_1 (sense, 5′-CCGAAGATTTCACAGTCAA-3′) and shRNA_2 (sense, 5′-CGGGGCGATCTCAAAAGTT-3′). The siRNA sequences used in this study were as follows and were synthesized by Sangon Biotech: Human *ELK4* siRNA_1 (sense, 5′-CCTCGAGTTTCCAGCGTGA-3′) and siRNA_2 (sense, 5′-CGACACAGACATTGATTCATT-3′), Human *SIRT7* siRNA_1 (sense, 5′-CCTGCCGTGTGAGGCGGAA-3′) and siRNA_2 (sense, 5′-GTGGACACTGCTTCAGAAA-3′).

### Transfection

HCT116 and SW620 cells were transfected with various plasmids, various packaged viruses, or siRNAs using Lipofectamine 2000 (Invitrogen) according to the manufacturer's instructions.

### Immunoprecipitation and immunoblotting

Proteins were extracted from cultured cells using a modified buffer (50 mM Tris-HCl (pH 7.4), 150 mM NaCl, 1 mM EDTA, 10% glycerol and 1% NP-40, protease inhibitor cocktail and phosphatase inhibitor cocktail), followed by immunoprecipitation and immunoblotting with the corresponding antibodies. Cell lysates were collected after centrifuged to remove the cell debris. For exogenous immunoprecipitation, cell lysates were incubated with flag-M2 beads (M8823, Sigma-Aldrich) at 4 °C overnight, and the beads were boiled after extensive washing. For endogenous immunoprecipitation, cell lysates were incubated with the indicated antibodies at 4 °C overnight, and the immunoprecipitate was incubated with protein A/G agarose beads (sc-2003, Santa Cruz Biotechnology) for 2–3 h followed by washing. Nuclear and cytosolic protein was prepared using the Nuclear and Cytoplasmic Protein Extraction Kit (Beyotime, P0028) according to the manufacturer's instructions. The protein concentration was determined using BCA Protein Assay Kit (Thermo Scientific). Protein samples were separated by SDS-PAGE, transferred onto PVDF membrane (Millipore), and probed with the indicated antibodies.

### Mass spectrometry analysis

Briefly, the immunoprecipitation sample of Flag-PYCR1 associated proteins was precipitated by acetone at −20 °C overnight and resuspended in 50 mM ammonium bicarbonate buffer containing Rapigest (Waters). The sample was heated at 95 °C for 10 min and 100 ng of sequencing-grade modified trypsin was added (Promega) before cooling down. The digestion was performed at 37 °C overnight and analysis was performed by LC-MS/MS using an Orbitrap-Elite mass spectrometer (Thermo Fisher Scientific). Proteins were identified by comparing the fragment spectra against those in the SWISS-PROT protein database (EBI) using Mascot v2.3 (Matrix Science) and Sequest v1.20 via Proteome Discoverer v1.3 (Thermo Fisher Scientific) software.

### Cell proliferation assay

Cells were incubated with CCK-8 (10% vol/vol) that was diluted into a normal culture medium at 37 °C for 1–4 h until the visual color conversion occurred. The samples were then analyzed by measuring the absorbance at 450 nm using Synergy H1 Microplate Reader (BioTek).

### Enzyme activity assay

All data were collected following the procedures previously described[25]. Enzyme activity assays were performed by monitoring the consumption of NADH (Ex. 350 nm, Em. 470 nm) using Synergy H1 Microplate Reader (BioTek). P5C was produced through the periodate oxidation of DL-5-hydroxylysine and was stable at 4 °C in 1 M HCl[26]. P5C was quantified with o-aminobenzaldehyde and adjusted to pH 7.5 with 1 M Tris-HCl (pH 9.0) immediately before using[27]. Assays were carried out at 37 °C for 5 min in the reaction buffer containing 0.1 M Tris-HCl

(pH 7.5), 0.01% Brij-35 detergent, 1 mM EDTA. Substrates and enzyme concentration were as follows: 60 nM PYCR1 enzyme, 3.5 mM P5C, and 0.5 mM NADH.

### Recombinant protein purification

DNA sequence encoding human PYCR1 was cloned into pET-28a (+) vector with the 6× His tag fused to N terminus. The PYCR1 construct was transformed into BL21(DE3)-competent *E. coli* cells and the cultures were grown at 37 °C to the absorbance at 600 nm of ~0.6 before inducing with IPTG (0.5 mM) at 16 °C overnight. Cells were lysed through sonicate after centrifuging and the supernatant was incubated with HisSeq Ni-NTA Agarose Resin for 2 h at 4 °C. Recombinant protein was eluted and concentrated using Ultrafree-15 centrifugal filters (Millipore).

### In vitro kinase assay

Bacterially purified recombinant WT or mutant His-PYCR1 was incubated with the tyrosine kinase domain of IGF1R (IGF1RK) in kinase assay buffer (20 mM $MgCl_2$, 0.1 mM DTT, 20 μg/mL BSA, and 100 mM HEPES at pH 7.5) supplemented with 1 mM ATP for 30 min at 2–3 °C. IGF1RK was removed by extensive washing after the reaction mixture incubating with HisSeq Ni-NTA Agarose Resin.

### P5CS activity assay

Briefly, the cells were incubated with hypotonic lysis buffer (10 mM HEPES (pH 7.9), 1.5 mM $MgCl_2$, and 10 mM KCl) for 10 min on ice. The buffer and cell volume ratio were 5:1. After gently resuspending the pellet three times, nuclei were collected by centrifugation at 4 °C for 5 min at 420 g, and then immediately subjected to immunoprecipitation. The P5CS protein was obtained by incubating 3× Flag-tag peptide with immunoprecipitates. The activity assay was performed using P5CS Activity Assay Kit (Solarbio, BC4425) according to the manufacturer's instructions.

### Proline measurement

The relative level of proline was measured using the Proline Content Assay Kit (Sangon Biotech, D799575). In brief, cells were homogenized in the extraction buffer and the supernatant was mixed with corresponding reagents following by incubation for 30 min at 100 °C. The values of absorbance were measured at 520 nm in a microplate reader.

### Library construction for RNA-seq and sequencing procedures

Total RNA was extracted from the cells using TRIzol Reagent according to the manufacturer's instructions (Invitrogen). RNA-seq transcriptome library was prepared following TruSeq™ RNA sample preparation Kit from Illumina (San Diego, CA) using 1 μg of total RNA. Shortly, messenger RNA was isolated according to polyA selection method by oligo (dT) beads and then fragmented by fragmentation buffer first. Secondly, double-stranded cDNA was synthesized using a SuperScript double-stranded cDNA synthesis kit (Invitrogen, CA) with random hexamer primers (Illumina). The synthesized cDNA was subjected to end-repair, phosphorylation, and 'A' base addition according to Illumina's library construction protocol. Libraries were size selected for cDNA target fragments of 300 bp on 2% Low Range Ultra Agarose followed by PCR amplified using Phusion DNA polymerase (NEB) for 15 cycles. After quantified by TBS380, the paired-end RNA-seq sequencing library was sequenced with the Illumina HiSeq X-ten/NovaSeq 6000 sequencer (2 × 150 bp read length). The library construction and sequencing were performed by Shanghai Majorbio Bio-pharm Technology Co., Ltd. The data were analyzed on the online platform of Majorbio Cloud Platform (www.majorbio.com).

### Real-time PCR analysis

Total RNA was extracted from cells using TRIzol (Invitrogen) according to the manufacturer's instructions. We synthesized cDNA from 1 μg total RNA using HyperScript III RT SuperMix for qPCR with gDNA Remover (EnzyArtisan, R202). The cDNA in quadruplicate were assessed for target mRNA level by quantitative real-time PCR with AceQ qPCR SYBR Green Master Mix (Q131-02). We calculated the relative mRNA level of target genes normalized to human β-actin mRNA level in the same samples. The qPCR primer sequences were as follows: h*KLK10*: 5′- GGA-GAGTGAAGTACAACAAGGG-3′ (forward) and 5′- CAGTCCAGCACA-TATCATGTTG-3′ (reverse); h*DEPP1*: 5′- GTCCACATAGACAGATGGACAG-3′ (forward) and 5′- AAAAGTCCAGCTTCTTAGGTCA-3′ (reverse); h*PTPRR*: 5′- AAATTGATATTCCGCGTCATGG-3′ (forward) and 5′- TAGGTGCTCAATGAATCGGTTA-3′ (reverse); h*CNN1*: 5′- GTGAACGTGG-GAGTGAAGTA-3′ (forward) and 5′- ATGATGTTCCGCCCTTCTCTTA-3′ (reverse); h*β-actin*: 5′- ACAATGTGGCCGAGGACTTTGA-3′ (forward) and 5′- TGTGTGGACTTGGGAGAGGACT-3′ (reverse).

### Chromatin immunoprecipitation assay

ChIP assay was performed using a Pierce Agarose ChIP Kit (Thermo, 26156) according to the manufacturer's guidelines. Quantitative real-time PCR was used to measure the amount of bound DNA, and the value of enrichment was calculated according to the percentage input method. Primers covering the ELK4-binding site of the *KLK10* and *DEPP1* genes promoter regions were used to perform real-time PCR. The following promoter-specific primers were used: h*KLK10* promoter: 5′- CCTGGAGTAAGGCTCTGTTTG-3′ (forward) and 5′- TGTGCAGGCTAACTGACTTCTTT-3′ (reverse); h*DEPP1* promoter: 5′- GAAGACACCCCTGACCACTG-3′ (forward) and 5′- TGAGGTGC-TAGTGTGTGCTG-3′ (reverse); h*GAPDH* promoter: 5′- GAGCCTC-GAGGAGAAGTTCC-3′ (forward) and 5′- GGACCCTTACACGCTTGGAT-3′ (reverse).

### Luciferase reporter assay

The *KLK10* promoter sequence (−1800bp to +200 bp) and *DEPP1* promoter sequence (−1800 bp to +200 bp) were inserted into the pGL3 vector and co-transfected with pRL-TK into the HEK293T cells. The luciferase activity was measured using the Dual Luciferase Reporter Gene Assay Kit (Beyotime, RG027). Data are presented as firefly/Renilla luciferase activity.

### Nuclear $NAD^+$ determination

The $NAD^+$ content was measured using the NAD(H) Content Assay Kit (Sangon Biotech, D799201). Specifically, cells were incubated with five times the volume of hypotonic lysis buffer (10 mM HEPES (pH 7.9), 1.5 mM $MgCl_2$, and 10 mM KCl) for 10 min on ice, and then nuclei were collected by centrifugation at 4 °C for 5 min at $420 \times g$. Extraction buffer was added to the nuclei and the supernatant was mixed with corresponding reagents according to the vendor's instructions. After centrifugation, the precipitation was dissolved and the data were recorded by measuring the absorbance at 570 nm.

### In vitro chromatin treatment

Isolation of chromatin extract from cultured cells was performed as described previously[28]. In summary, cells were washed with prechilled PBS and resuspended in solution A (10 mM HEPES (pH 7.9), 0.1% Triton X-100, 10 mM KCl, 1.5 mM MgCl2, 0.34 M sucrose, 10% glycerol, 1 mM DTT, 10 mM NaF, 1 mM Na2VO3, and protease inhibitors). The mixture was incubated for 5 min on ice, and followed by centrifugation at a speed of $1300 \times g$ for 4 min at 4 °C. The precipitate was lysed in solution B (3 mM EDTA, 0.2 mM EGTA, 1 mM DTT, and protease inhibitors). Insoluble chromatin was collected after centrifugation. The chromatin extract was resuspended in reaction buffer (20 mM Tris-HCl (pH 7.9), 100 mM KCl, 5 mM $MgCl_2$, 0.2 mM EDTA, 10% glycerol, 0.5 mM DTT) in the presence or absence of indicated metabolite at 30 °C for 1 h. The chromatin extract was fixed and sonicated, and ChIP-qPCR analysis with H3K18ac antibody was performed using a Pierce Agarose ChIP Kit.

## Immunofluorescence analysis

Cells were seeded on cell slides at a suitable density and then cultured under the indicated condition. Cells were fixed and incubated with primary antibody, Alexa Fluor dye-conjugated secondary antibodies, and DAPI according to standard protocols. The images were captured using a confocal microscope (Olympus IX83).

## Immunohistochemical analysis

Formalin-fixed, paraffin-embedded consecutive human colon cancer tissue sections (3–5 mm) were deparaffinized and rehydrated. Antigen retrieval was performed by boiling tissue sections in 10 mM citrate buffer (pH 6.0) in a microwave oven for 5 min. The activity of endogenous peroxidase was blocked with 3% hydrogen peroxide in methanol for 10 min at room temperature. After washing, nonspecific binding sites were blocked by incubating the slides with 10% FBS/PBS for 30 min at room temperature. Sections were subsequently incubated with antibodies, at 4 °C overnight. After incubation with the primary antibodies, the sections were washed and incubated with secondary antibodies and DAB staining reagent with GTVisionTM Detection System/Mo&Rb Kit according to manufacturer's instructions. After counterstaining with hematoxylin and dehydration, the sections were mounted and imaged using the Leica microscope. Immunoreactivity was semiquantitatively evaluated according to intensity and area: the staining intensity was graded on a scale from 0 to 3; the percentage of immunoreactivity was scored on a scale from 0 to 3. These numbers were then multiplied resulting in a score of 0-9. Statistical significance was set at $p < 0.05$. Human tumor samples and their paired noncancerous tissues were obtained from CRC patients with surgery. Patients with radiotherapy or chemotherapy treatment before surgery were excluded. Survival time was calculated from the date of surgery to the date of death or last follow-up. Written informed consent was obtained from each patient and the investigation was approved by the institutional review board of Shanghai East Hospital, Tongji University, Shanghai, China. The tumor-node-metastasis staging was performed according to American Joint Committee on Cancer (AJCC) standards.

## Statistics and reproducibility

Statistical testing was performed using the two-tailed Student's $t$ test, Pearson's chi-squared test, or Pearson correlation analysis, and a $P$ value of 0.05 was considered as a borderline for statistical significance. Excel (V2019, Microsoft, Redmont, WA) was used for data collection. Data analysis was performed utilizing SPSS Statistics 20 (SPSS, Chicago, IL, USA). All experiments were performed at least three times unless otherwise indicated, and representative results were shown in the figures.

## Reporting summary

Further information on research design is available in the Nature Portfolio Reporting Summary linked to this article.

# Data availability

RNA-sequencing datasets are available on Genome Sequence Archive under accession number HRA003999 under controlled access. The GSA data is only accessible for principal investigators and there is no time limit for access to be granted. Principal Investigators can register and request the datasets or email Ke Zheng at zhengke1215@163.com Source data are provided with this paper.

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

## Acknowledgements

This work was supported by grants from National Nature Science Foundation of China (grant no. 82073224, 81972586, 81773006, 82072684), Shanghai Committee of Science and Technology (grant no. 20XD1402900, 21XD1423100, 21JC1405200), National Key R&D Program of China (grant no. 2020YFA0803602), Shanghai Municipal Health Commission (grant no. 20224Z0016) and the Medical Discipline Construction Project of Pudong Health Committee of Shanghai (grant no. PWZxq2022-6, PWYgf2021-2).

## Author contributions

T.C. and Y.J. designed the study; K.Z. and N.S. performed the experiments; T.C., Y.J., M.X., K.Z., G.H., Z.L., Q.Z., and L.Z. analyzed and interpreted the data; T.C. and Y.J. wrote and revised the manuscript. Q.Z., G.H., and L.H. reviewed the manuscript and gave conceptual advice; T.C., Y.J., and M.X. provided technical and material support. All authors approved the manuscript.

## Competing interests

The authors declare no competing interests.
