## [Peer Review File · Nature Communications]

IGF1R-phosphorylated PYCR1 facilitates ELK4 transcriptional activity and sustains tumor growth under hypoxiaREVIEWER COMMENTS

Reviewer #1 (Remarks to the Author):

I carefully reviewed the manuscript, #NCOMMS-23-01048, entitled IGF1R-phosphorylated PYCR1 facilitates ELK4 transcriptional activity and sustains tumor growth under hypoxia. In this manuscript, the authors showed that hypoxia promotes phosphorylation of PYCR1 by IGF1R in the nucleus, and that this phosphorylation triggers the complex formation between PYCR1 and ELK4 and subsequently the recruitment of them to the ELK4 target gene promoter. Based on its binding to ELK4, the NADH oxidase activity of PYCR1 promotes ELK4-Sirt7-mediated gene transcriptional repression and prevents cell growth arrest under hypoxia, where NAD⁺ generation by PYCR1 has been found to locally enhance the deacetylation activity of Sirt7 that is essential for transcriptional repression. Moreover, their physiological analyses demonstrate the important relevance of the compartmental regulation by PYCR1 to CRC development.

I think the manuscript presented important aspect of tumor growth under hypoxia but there are some weak points in the design of experiments. Approaching the problems and moreover validating their findings not only in vitro but also in vivo will further strengthen this manuscript and justify its publication in Nature Communications.

Major comments:

1. Figure 1F: I'd like to ask the authors to examine whether nuclear localization of PYCR1 depends on the complex formation with IGF1R and/or ELK4 and on the phosphorylation of PYCR1 by IGF1R. I'm curious if the PYCR nuclear localization would be influenced by silencing IGF1R and/or ELK4 or by introducing a point mutation into PYCR1 Y135. They are informative to readers of the paper.
2. Figure 1I: I'd like to ask the authors to examine the impact of PYCR1 silencing on cell growth under normoxia, too.
3. It would be necessary to examine whether the nuclear localization of PYCR1 is not affected by the intratumor oxygen environment (based on the results in Fig 1E) by detecting hypoxic regions with endogenous hypoxic markers in case of clinical tumor specimens and with endogenous markers or an exogenous marker, pimonidazole, in case of xenografted tumors. Based on the results of Fig 2, phosphorylation of PYCR1 is expected to be increased in the hypoxic regions of tumor tissues; this should be also tested.
4. The authors indicated, based on the results in Figure 1, that nuclear localization of PYCR1 is required for sustained cell growth under hypoxia, which is independent on its metabolic activity for proline synthesis. But in order to further make the conclusion solid, it would be critical to carry out the same experiments as Figure 1I under normoxia as well.
5. The authors need to examine whether the potential ELK4-binding consensus sequence (CCGGA or TTCC) actually works or not by introducing mutation into them.

Minor comments:

1. The authors carried out experiments only under hypoxia in Figs 1C, 1G, 2A, 2B, 3E, 3F, 4D, 4E, 4F, 5A, 5B, 5F, and 5G, but are expected to conduct them under normoxia, too.
2. Figure 1G: The cell lysates should be analyzed by not only antibodies for phosphorylated forms of IGF1R, AKT, and ERK but also their pan-antibodies.

Reviewer #2 (Remarks to the Author):

In the manuscript by Zheng et al the authors study the role of PYCR1, an enzyme involved in proline synthesis, in tumor growth under hypoxia. The involvement of proline synthesis in tumorigenesis and its role in hypoxia has been previously described but not fully characterized. Previous studies identified PYCR1 as a major regulator of tumor growth and survival under these conditions by regulation of TCA cycle. The authors show that PYCR1, the transcription factor ELK4 and IGFR1 kinase form a complex upon hypoxia. IGFR1 phosphorylates Y135 in PYCR1, which promotes its binding to ELK4, and regulates the expression of a subset of hypoxia-related genes through a role independent of proline biosynthesis. ELK4 was previously shown to interact with the NAD⁺-dependent sirtuin SIRT7 to repress ELK4-dependent genes. The authors propose that the role of PYCR in ELK4-dependent hypoxia response is to boost local SIRT7 activity through the production of NAD⁺. The authors link this mechanism to colon cancer tumors, where PYCR1 is found by IHC to localize also in the nucleus.

Understanding the mechanisms and pathways that regulate hypoxia and how their alteration may contribute to sustain tumor survival and growth is very important and of high interest to a wide range of researchers from different disciplines. The authors have done a good job with the identification of the IGFR1-PYCR-ELK4-SIRT7 regulatory axis in hypoxia response. The work is technically well done, the findings are quite relevant and the results are compelling. However, there are some major issues that should be addressed by the authors:

1) A major issue of the work is the model by which PYCR1 regulates the expression of ELK4-dependent genes. The authors propose that PYCR1 gets recruited to these genes to increase the local concentration of NAD⁺ and therefore induce SIRT7 activity. This would fit with the reduced NAD/NADH ratio reported by some studies, but the model needs to be validated with more evidence:

a) The observation that PYCR1 pY135 is exclusively nuclear is striking as PYCR1 is a mitochondrial enzyme. However, these studies are based only on cellular fractionation. In the IHCs of cancer samples, it's difficult to assess this, as it seems to be everywhere in the cell. In some cases, depending on how the cell fractionation is done, nucleus and mitochondria could be pulled down together and separated from the cytoplasmic fraction.

Can the authors discard that these nuclear fractions do not contain mitochondria? A mitochondrial protein control in the western blots would solve this. Additionally, the authors should further confirm the nuclear localization in these cell lines by other methods like IF.

b) If the authors are correct, the overexpression of the mN1 mutant of figure 1 should not have any impact on the expression of these ELK4-dependent genes or the H3K18ac enrichment. Is that the case?

c) The effect of downregulation of mitochondrial P5CS observed on the role of PYCR1 in the nucleus (Figure 5D-E) is interesting. The authors assume that it increases glutamate 5-semialdehyde (G5S), but does this metabolite ever reach the nucleus? Or is P5CS also in the nucleus? The authors should provide evidence in this sense.

d) Linked to this, would overexpression of the addition of G5S combined with NADH overcome the effect of PYCR1 downregulation on SIRT7 activity?

e) Similarly, the results with NMNAT1 overexpression are very interesting, but in some somehow unclear. If the only impact of PYCR1 is to produce NAD for SIRT7 upon hypoxia, and NMNAT1 overexpression can revert the effect of PYCR1 WT mutant overexpression (in figure 5), can NMNAT1 overexpression compensate the impact of PYCR1 downregulation on hypoxia response?

f) Does SIRT7 interact directly with PYCR1? The IPs in figure 4 by ELK4 only show that ELK4 interacts with both but not that PYCR1 and SIRT7 interact directly.

2) The authors perform all the genomic and epigenetic validations on four target genes of

ELK4, SIRT7, and PYCR1. However, they do not show any negative control to demonstrate that the impact of PYCR on for instance H3K18ac only takes place in these ELK4 genes.

3) If the model is correct there should be a nuclear increase in the levels of Proline in the nucleus. Did the authors test this possibility? Alternatively, did the authors consider the possibility that PYCR1 may modify chemically SIRT7?

4) Recent studies have suggested that H3K36ac is a better SIRT7 substrate than H3K18ac. Is H3K36ac also altered in these target genes as K18ac?

Minor points:

1) The label of the IPs is a little bit confusing, specifically when the authors include in the upper left part of the experiment "IgG" by itself without any other label. In the sake of clarity the authors should clearly state close to this "IgG" label the antibody used in the IP.

Reviewer #3 (Remarks to the Author):

In this study, the authors show that pyrroline-5-carboxylate reductase-1 (PYCR1) is constitutively localized in the nucleus of colorectal cancer cells. In hypoxia conditions, IGF1R phosphorylates PYCR1 at Tyrosine135 and promotes the binding of PYCR1 with ETS-like transcription factor 4 (ELK4). Mechanistically, this complex is key to suppressing the expression of genes associated with cell growth arrest. Specifically, the PYCR1 reaction generates NAD⁺, which supports the deacetylase activity of Sirt7 and, in turn, gene transcription suppression by ELK4. Notably, the authors were able to associate this specific PYCR1 phosphorylation with colorectal cancer growth.

Overall, the manuscript presents an interesting and novel mechanism for the PYCR1 role and its capacity to promote cancer cell growth under hypoxia. The biochemistry studies include the proper controls and thoroughly show the mechanistic link between IGF1R, PYCR1, Sirt7, and ELK4. Moreover, the therapeutic link is also present. In conclusion, I encourage the publication of this study in Nature Communications. I only have a few comments that I believe will strengthen even more the main conclusions.

Major:

1. In Figure 1B, the authors show that PYCR1 is localized in the nucleus also in normoxia. Does nuclear PYCR1 interact with other genes in normoxia? A better characterization of PYCR1 interaction in these conditions will help strengthen its specific role in hypoxia. The mechanistic characterization of PYCR1 interaction in normoxia is not required.

2. All the in vitro studies of this manuscript were performed in 1% oxygen. Do the authors observe a different response of tumor cells in vivo depending on the region in which they are localized (normoxic regions vs hypoxic regions)?

3. P5CS is normally located in the mitochondria. Does the product of the P5CS reaction diffuse to the cytosol and then to the nucleus? Did the authors find P5CS in the nucleus? More experiments elucidating the compartmentalization of proline biosynthesis are needed.

Minor:

1. Some of the sentences in the manuscript are oddly articulated and thus difficult to understand. I suggest revising the manuscript so that it is easier to follow by the readers.

Reviewer #1 (Remarks to the Author):

I carefully reviewed the manuscript, #NCOMMS-23-01048, entitled IGF1R-phosphorylated PYCR1 facilitates ELK4 transcriptional activity and sustains tumor growth under hypoxia.

In this manuscript, the authors showed that hypoxia promotes phosphorylation of PYCR1 by IGF1R in the nucleus, and that this phosphorylation triggers the complex formation between PYCR1 and ELK4 and subsequently the recruitment of them to the ELK4 target gene promoter. Based on its binding to ELK4, the NADH oxidase activity of PYCR1 promotes ELK4-Sirt7-mediated gene transcriptional repression and prevents cell growth arrest under hypoxia, where NAD⁺ generation by PYCR1 has been found to locally enhance the deacetylation activity of Sirt7 that is essential for transcriptional repression. Moreover, their physiological analyses demonstrate the important relevance of the compartmental regulation by PYCR1 to CRC development. I think the manuscript presented important aspect of tumor growth under hypoxia but there are some weak points in the design of experiments. Approaching the problems and moreover validating their findings not only in vitro but also in vivo will further strengthen this manuscript and justify its publication in Nature Communications.

Answer: We greatly appreciate the reviewer's acknowledgement of the significance of the present study and the insightful comments that will definitely strengthen our manuscript.

Major comments:

1. Figure 1F: I'd like to ask the authors to examine whether nuclear localization of PYCR1 depends on the complex formation with IGF1R and/or ELK4 and on the phosphorylation of PYCR1 by IGF1R. I'm curious if the PYCR nuclear localization would be influenced by silencing IGF1R and/or ELK4 or by introducing a point mutation into PYCR1 Y135. They are informative to readers of the paper.

Answer: Thanks for pointing out this important issue. In the revised manuscript, we performed new experiments according to reviewer's suggestion. The nuclear level of PYCR1 was examined under ELK4- and IGF1R depletion, or under expression of rPYCR1 Y135F, and compared with their counterpart respectively. As a result, neither IGF1R- nor ELK4 silencing affect the nuclear level of PYCR1 (Supplementary Fig. 1k). Meanwhile, cellular fractionation analysis showed PYCR1 Y135F displayed a comparable nuclear level to that of WT PYCR1 (Supplementary Fig. 2i). These results indicate IGF1R-promoted binding of PYCR1 to ELK4 is dispensable for PYCR1 nuclear localization. In addition, immunofluorescence analysis also verified that IGF1R depletion did not affect cellular distribution of PYCR1 (Supplementary Fig. 2f, bottom panel).

2. Figure 1I: I'd like to ask the authors to examine the impact of PYCR1 silencing on cell growth under normoxia, too.

Answer: Thanks for pointing out this important issue. In the revised manuscript, we performed new experiments according to reviewer's suggestion, in which the effect of

PYCR1 depletion or expression of rPYCR1 mN1 on cell growth was further examined under normoxia respectively. As shown in (Supplementary Fig. 1n and 1o) of revision, PYCR1 depletion resulted in an impairment of cell growth under normoxia, although the extent of which is shown to be lesser than that under hypoxia. Along with the results from expression of rPYCR1 Y13F and rPYCR1 mN1 (Fig. 1i and Supplementary Fig. 1m), these results reveal the elevated impact of PYCR1 on cell growth under hypoxia, which is in line with the observation previously reported [Cell Rep. 38(5):110320].

3. It would be necessary to examine whether the nuclear localization of PYCR1 is not affected by the intratumor oxygen environment (based on the results in Fig 1E) by detecting hypoxic regions with endogenous hypoxic markers in case of clinical tumor specimens and with endogenous markers or an exogenous marker, pimonidazole, in case of xenografted tumors. Based on the results of Fig 2, phosphorylation of PYCR1 is expected to be increased in the hypoxic regions of tumor tissues; this should be also tested.

Answer: This question is well taken. According to reviewer's suggestion, we analyzed nuclear level and phosphorylation level of PYCR1 and in clinical tumor specimens and distinct regions of xenografted tumors lysates. As a result, the intensity of PYCR1 pY135 (Supplementary Fig. 6g) rather than the total level of PYCR1 (Supplementary Fig. 1a) correlates HIF-1 α levels collected clinical tumor specimens. Meanwhile, in the distinct regions of xenografted tumors with higher level of HIF-1 α , PYCR1 pY135 was generally detected with a stronger signals, while which was not occurred in the total protein level of PYCR1 (Supplementary Fig. 6e). These data indicate nuclear localization of PYCR1 is localized in the nucleus regardless of intratumor oxygen condition, and PYCR1 phosphorylation is induced under hypoxia.

4. The authors indicated, based on the results in Figure 1, that nuclear localization of PYCR1 is required for sustained cell growth under hypoxia, which is independent on its metabolic activity for proline synthesis. But in order to further make the conclusion solid, it would be critical to carry out the same experiments as Figure 1I under normoxia as well.

Answer: Thanks for pointing out this important issue. In the revised manuscript, we performed new experiments to examine the effect of exogenous proline on cell growth under expression of rPYCR1 mN1 either at normoxia or hypoxia condition. It was found that supplementation of exogenous proline did not affect cell growth significantly regardless of oxygen condition in rPYCR1 mN1-expressing cells (Fig. 1i and Supplementary Fig. 1m).

5. The authors need to examine whether the potential ELK4-binding consensus sequence (CCGGA or TTCC) actually works or not by introducing mutation into them.

Answer: Thanks for pointing out this issue. In the revised manuscript, the consensus sequence-binding ability and transcriptional effect of ELK4 on the target genes was validated by the luciferase analysis. As a result, ELK4 overexpression soundly

promoted the reporter activity of WT promoters (from genes *KLK10* and *DEPP1*), and this effect was abolished in the promoters with mutation in the ELK4-binding consensus sequence (Supplementary Fig. 3b). The promoting-effect of ELK4 overexpression on luciferase expression suggest the transcriptional activity of ELK4 in the genomic region would be differently exhibited from that shown in the luciferase report assay.

Minor comments:

1. The authors carried out experiments only under hypoxia in Figs 1C, 1G, 2A, 2B, 3E, 3F, 4D, 4E, 4F, 5A, 5B, 5F, and 5G, but are expected to conduct them under normoxia, too.

Answer: Thanks for pointing out these issues. For the according results as indicated by the reviewer, we performed new experiments that includes normoxia condition. Collectively, these results respectively indicated N1 sequence of PYCR1 was required for its nuclear accumulation either under normoxia or hypoxia (Fig. 1c and 1i). PYCR1 specifically interacted with ELK4 under hypoxia (Fig. 1g and 2a, Supplementary Fig. 1j and 2a); The enrichment of PYCR1 (Fig. 3e and 3f) and SIRT7 (Fig. 4d) on ELK4-targeted genes promoter is largely induced under hypoxia; the promoting-effect of SIRT7 depletion and expression of rPYCR1s on ELK4-targeted gene transcription were specifically detected under hypoxia (Fig. 4f and 4g) but not under normoxia (Supplementary Fig. 4e and 4f); and the rescue-effect of NMNAT-1 overexpression on ELK4-targeted gene transcription and promoter-associated level of H3K18ac were restricted in rPYCR1 Y135F- or rPYCR1 T238A-expressing cells under hypoxia (Fig. 5a and 5b) instead of normoxia (Revision Fig. 1 and 2); the according effect of P5C/NADH and NAD⁺ on H3K18ac at ELK4-targeted genes promoter was only detected in chromatin extracts collected from cells cultured under hypoxia (Fig. 5f and 5g) rather than normoxia (Revision Fig. 3).

Revision Fig. 1

Revision Fig. 1. HCT116 cells expressing the indicated Flag-rPYCR1s were simultaneously overexpressed with or without NMNAT1. Cells were cultured under normoxia and supplemented with the metabolites as indicated. mRNA levels of indicated ELK4-targeted genes were analyzed

by real-time PCR. data are presented as the mean \pm s.d.; statistical analysis was performed using the two-tailed Student's *t*-test.

Revision Fig. 2

Revision Fig. 2. HCT116 cells expressing the indicated Flag-rPYCR1s were simultaneously overexpressed with or without NMNAT1. Cells were cultured under normoxia and supplemented with the metabolites as indicated. ChIP analyses were performed with indicated antibodies, the primers covering the ELK4-binding sites at promoter regions of the indicated genes were utilized for real-time PCR analysis. The y axis shows the value normalized to the input. data are presented as the mean \pm s.d.; statistical analysis was performed using the two-tailed Student's *t*-test.

Revision Fig. 3

Revision Fig. 3. HCT116 cells with depleted P5CS (a) or with expression of rPYCR1 T238A (b) were transfected with or without Sirt7 siRNA. Cells were cultured under normoxia. Chromatin

extracts were collected and mixed with or without the indicated metabolites for 30 min. ChIP analyses were performed with indicated antibodies, the primers covering the ELK4-binding sites at promoter regions of the indicated genes were utilized for real-time PCR analysis. The y axis shows the value normalized to the input. Data are presented as the mean \pm s.d.; statistical analysis was performed using the two-tailed Student's *t*-test.

2. Figure 1G: The cell lysates should be analyzed by not only antibodies for phosphorylated forms of IGF1R, AKT, and ERK but also their pan-antibodies.

Answer: Thanks for pointing out this issue. The original data including total level of IGF1R, AKT, and ERK previously was examined using identical cell lysates and the according data was added into the revision (Figure 1g and Supplementary Fig. 1j).

Reviewer #2 (Remarks to the Author):

In the manuscript by Zheng et al the authors study the role of PYCR1, an enzyme involved in proline synthesis, in tumor growth under hypoxia. The involvement of proline synthesis in tumorigenesis and its role in hypoxia has been previously described but not fully characterized. Previous studies identified PYCR1 as a major regulator of tumor growth and survival under these conditions by regulation of TCA cycle. The authors show that PYCR1, the transcription factor ELK4 and IGFR1 kinase form a complex upon hypoxia. IGFR1 phosphorylates Y135 in PYCR1, which promotes its binding to ELK4, and regulates the expression of a subset of hypoxia-related genes through a role independent of proline biosynthesis. ELK4 was previously shown to interact with the NAD⁺-dependent sirtuin SIRT7 to repress ELK4-dependent genes. The authors propose that the role of PYCR in ELK4-dependent hypoxia response is to boost local SIRT7 activity through the production of NAD⁺. The authors link this mechanism to colon cancer tumors, where PYCR1 is found by IHC to localize also in the nucleus.

Understanding the mechanisms and pathways that regulate hypoxia and how their alteration may contribute to sustain tumor survival and growth is very important and of high interest to a wide range of researchers from different disciplines. The authors have done a good job with the identification of the IGFR1-PYCR-ELK4-SIRT7 regulatory axis in hypoxia response. The work is technically well done, the findings are quite relevant and the results are compelling. However, there are some major issues that should be addressed by the authors:

Answer: We greatly appreciate the reviewer's acknowledgement of the significance of the present study and the insightful comments that will definitely strengthen our manuscript.

1) A major issue of the work is the model by which PYCR1 regulates the expression of ELK4-dependent genes. The authors propose that PYCR1 gets recruited to these genes to increase the local concentration of NAD⁺ and therefore induce SIRT7 activity. This would fit with the reduced NAD/NADH ratio reported by some studies, but the model needs to be validated with more evidence:

a) The observation that PYCR1 pY135 is exclusively nuclear is striking as PYCR1 is a mitochondrial enzyme. However, these studies are based only on cellular fractionation. In the IHCs of cancer samples, it's difficult to assess this, as it seems to be everywhere in the cell. In some cases, depending on how the cell fractionation is done, nucleus and mitochondria could be pulled down together and separated from the cytoplasmic fraction. Can the authors discard that these nuclear fractions do not contain mitochondria? A mitochondrial protein control in the western blots would solve this. Additionally, the authors should further confirm the nuclear localization in these cell lines by other methods like IF.

Answer: Thanks for pointing out this important issue. In the revised manuscript, we conducted additional analyses in terms of cellular distribution of PYCR1 pY135. In line with the result of Fig. 1e showing IGF1R-PYCR1 interaction was specifically

detected in the nucleus, both cellular fraction (Supplementary Fig. 2d) and immunofluorescence (Supplementary Fig. 2e) analysis indicated the nuclear accumulation of PYCR1 pY135 rather than total PYCR1 was largely compromised in rPYCR1 mN1-expressing cells, in which COX IV was used as a mitochondrial protein control. Meanwhile, and the requirement of IGF1R for nuclear accumulation of PYCR1 pY135 under hypoxia was also validated by the immunofluorescence analysis (Supplementary Fig. 2f). Further analysis showed that the amount of PYCR1 pY135 accounted for around 35% of total nuclear level of PYCR1 under hypoxia (Supplementary Fig. 2g, left panel). In contrast, the amount of mitochondrial PYCR1 was unaffected by depletion of PYCR1 pY135 using the according antibody (Supplementary Fig. 2g, right panel).

b) If the authors are correct, the overexpression of the mN1 mutant of figure 1 should not have any impact on the expression of these ELK4-dependent genes or the H3K18ac enrichment. Is that the case?

Answer: This question is well taken and we performed new experiment according to reviewer's suggestion. The original results indicated nuclear interaction between PYCR1 and ELK4 (mediated by PYCR1 pY135) was required for transcriptional repression of ELK4-targeted genes under hypoxia (Fig. 3b). In line with this, under hypoxia reconstituted expression of rPYCR1 mN1 lost the suppressive effect on expression of ELK4-dependent genes (Supplementary Fig. 3c) and H3K18ac enrichment (Revision Fig. 4), as indicated by the elevated genes transcription compared with that of WT rPYCR1.

Revision Fig. 4

Revision Fig. 4. HCT116 cells with depletion of endogenous PYCR1 were reconstitutively expressed with WT rPYCR1 or rPYCR1 mN1. Cells were cultured under normoxia or hypoxia for 12 h. ChIP analyses were performed with indicated antibodies, the primers covering the ELK4-binding sites at promoter regions of the indicated genes were utilized for real-time PCR analysis. The y axis shows the value normalized to the input. data are presented as the mean \pm s.d.; statistical analysis was performed using the two-tailed Student's *t*-test.

c) The effect of downregulation of mitochondrial P5CS observed on the role of PYCR1 in the nucleus (Figure 5D-E) is interesting. The authors assume that it increases glutamate 5-semialdehyde (G5S), but does this metabolite ever reach the nucleus? Or is P5CS also in the nucleus? The authors should provide evidence in this sense.

Answer: Thanks for pointing out this important issue and we performed new experiment according to reviewer's suggestion. As a result, P5CS was also detected in the nucleus, and its level (Supplementary Fig. 5h) as well as the metabolic activity (Supplementary Fig. 5i) was even increased to certain degree under hypoxia. In the meantime, metabolite of nuclear fraction (Please see in the section of "Method") further showed the nuclear amount of proline was increased under hypoxia, which was largely decreased by P5CS depletion (Supplementary Fig. 5f). Similarly, the overall level of proline level (Fig. 2j) as well as the nuclear amount of proline (Supplementary Fig. 2o) was augmented under hypoxia in WT rPYCR1- or rPYCR1 Y135F-expressing cells, which was blocked in rPYCR1 T238A-expressing cells. These data reveal an enhancement of the sequential reactions for proline synthesis at this condition, and the indispensable role of P5CS in maintaining the nuclear level of proline. In terms of measurement of P5C amount, the pretest from mass spectrometry platforms we tried to contact revealed it is unable to stably and precisely proceed the experiment, which would be due to the chemical properties of P5C. Here we hope this issue could be understood and believe the technical improvement in this regard will be definitely beneficial for the further study.

d) Linked to this, would overexpression of the addition of G5S combined with NADH overcome the effect of PYCR1 downregulation on SIRT7 activity?

Answer: This question was well taken and we performed new experiment according to reviewer's suggestion. As a result, promoter-associated accumulation of H3K18Ac induced upon P5CS- (Fig. 5f) or PYCR1-depletion (Supplementary Fig. 5j) under hypoxia was successfully deacetylated by addition of NAD⁺ in a SIRT7 dependent manner. Of note, P5CS-depletion failed to increased promoter-associated accumulation of H3K18Ac cells after culturing under normoxia (Revision Fig. 3). In contrast with that in P5CS-depleted cells, the deacetylation effect from incubation of P5C combined with NADH was not while only NAD⁺ treatment was detected in PYCR1-depleted cells (Supplementary Fig. 5j). Consistently, for chromatins collected from rPYCR1 T238A-expressing cells, the intensity of H3K18Ac at promoter region was readily removed with NAD⁺ incubation in a SIRT7 dependent manner (Fig. 5g).

Revision Fig. 3

Revision Fig. 3. HCT116 cells with depleted P5CS (a) or with expression of rPYCR1 T238A (b) were transfected with or without SIRT7 siRNA. Cells were cultured under normoxia. Chromatin extracts were collected and mixed with or without the indicated metabolites for 30 min. ChIP analyses were performed with indicated antibodies, the primers covering the ELK4-binding sites at promoter regions of the indicated genes were utilized for real-time PCR analysis. The y axis shows the value normalized to the input. Data are presented as the mean \pm s.d.; statistical analysis was performed using the two-tailed Student's *t*-test.

e) Similarly, the results with NMNAT1 overexpression are very interesting, but in some somehow unclear. If the only impact of PYCR1 is to produce NAD for SIRT7 upon hypoxia, and NMNAT1 overexpression can revert the effect of PYCR1 WT mutant overexpression (in figure 5), can NMNAT1 overexpression compensate the impact of PYCR1 downregulation on hypoxia response?

Answer: This question was well taken and we performed new experiment according to reviewer's suggestion. As a result, overexpression of NMNAT-1 (Supplementary Fig. 5a), which substantially increased nuclear level of NAD⁺ (Supplementary Fig. 5b) under hypoxia, partially reversed genes transcription and promoter-associated H3K18Ac accumulation in rPYCR1 Y135F- or rPYCR1 T238A-expressing cells (Fig. 5a and 5b) as well as in PYCR1-depleted cells (Supplementary Figs. 5c and 5d), suggesting NAD⁺ instead of proline availability is importantly involved in genes transcription regulation by PYCR1. It could be assumed mutation of PYCR1 Y135F uncouples the proximal metabolic activity of PYCR1 for ELK4-mediated transcriptional activity and thereby resulted in the change of genes expression, while

NMNAT-1 overexpression produces a compensative effect in this regard.

f) Does SIRT7 interact directly with PYCR1? The IPs in figure 4 by ELK4 only show that ELK4 interacts with both but not that PYCR1 and SIRT7 interact directly.

Answer: This question was well taken. In the revised manuscript, we performed further analysis to examine whether SIRT7 interact directly with PYCR1. As shown in Supplementary Fig. 4b, the binding of PYCR1 to SIRT7 was increased under hypoxia, which was found to be blocked upon ELK4 depletion, revealing the PYCR1 interacts with SIRT7 indirectly in a ELK4 dependent manner. In addition, the original data in Fig. 4a had shown the binding of PYCR1 pY135 is dispensable for the interaction between ELK4 and SIRT7. These data reveal the central role of ELK4 in PYCR1-ELK4-SIRT7 complex formation.

2) The authors perform all the genomic and epigenetic validations on four target genes of ELK4, SIRT7, and PYCR1. However, they do not show any negative control to demonstrate that the impact of PYCR on for instance H3K18ac only takes place in these ELK4 genes.

Answer: This question was well taken. In the revised manuscript, we performed ChIP analysis to examine the accumulation of PYCR1 and H3K18 acetylation at *GAPDH* promoter region that does not contain ELK4 binding-consensus sequence. As a result, *GAPDH* promoter-associated level of PYCR1 (Supplementary Fig. 3f) and H3K18 acetylation (Supplementary Fig. 4h) was shown to be limited, which was indistinguishable among according experimental groups. These data validate the role of *GAPDH* gene as a negative control to examine the effect of PYCR1 on the regulation of relevant epigenetic modification.

3) If the model is correct there should be a nuclear increase in the levels of Proline in the nucleus. Did the authors test this possibility? Alternatively, did the authors consider the possibility that PYCR1 may modify chemically SIRT7?

Answer: This question was well taken and we performed new experiment according to reviewer's suggestion. Similar to the overall level of proline level (Fig.2j), the nuclear level of proline was augmented under hypoxia in WT rPYCR1- or rPYCR1 Y135F-expressing cells (Supplementary Fig. 2o), which was blocked in rPYCR1 T238A-expressing cells. In addition, P5CS was found to be localized in the nucleus, and its level (Supplementary Fig. 5h) as well as the metabolic activity (Supplementary Fig. 5i) was even increased to certain degree under hypoxia. In the meantime, metabolites of nuclear fraction analysis (Please see in the section of "Method") further showed P5CS depletion largely decreased the nuclear amount of proline under hypoxia (Supplementary Fig. 5f). These data reveal an enhancement of the sequential reactions for proline synthesis at this condition.

Moreover, results shown in Fig. 5f indicated expression of addition of combined P5C and NADH facilitated promoter-associated H3K18 deacetylation in vitro, while this effect was blocked by SIRT7 silencing, suggesting the essential role of SIRT7. The in vitro assay also showed indicated addition NAD⁺ but not P5C-NADH incubation was

able to remove promoter-associated H3K18 deacetylation in PYCR1-depleted cells (Supplementary Fig. 5j), which was inhibited by SIRT7 silencing. These data further demonstrate the activity of PYCR1 for NAD⁺ production is required for SIRT7-mediated H3K18 deacetylation at promoter of ELK4-targeted genes.

4) Recent studies have suggested that H3K36ac is a better SIRT7 substrate than H3K18ac. Is H3K36ac also altered in these target genes as K18ac?

Answer: This question was well taken and we performed new experiment according to reviewer's suggestion. As shown in the revised data, it was found that the promoter-associated level of H3K36Ac (H3K36 acetylation), an alternative substrate of SIRT7 known to potentially mark active gene promoters [J Am Chem Soc. 141(6):2462-2473], was also increased by Sirt7 depletion (Supplementary Fig. 4i, left panel) or rPYCR1 Y135F (or rPYCR1 T238A) expression (Supplementary Fig. 4j, left panel) for ELK4-targeted gene *KLK10*. For *DEPPI*, a limited effect of SIRT7 depletion (Supplementary Fig. 4i, right panel) and PYCR1 mutants (Supplementary Fig. 4j, right panel) on the level of promoter-associated H3K36 acetylation was detected, which would be due to the restricted activity for H3K36 acetylation at *DEPPI* promoter.

Minor points:

1) The label of the IPs is a little bit confusing, specifically when the authors include in the upper left part of the experiment "IgG" by itself without any other label. In the sake of clarity the authors should clearly state close to this "IgG" label the antibody used in the IP.

Answer: Thanks for pointing out the issue. In the revised manuscript, we have made the modification according to reviewer's suggestion, in which the label of the IPs was reorganized.

Reviewer #3 (Remarks to the Author):

In this study, the authors show that pyrroline-5-carboxylate reductase-1 (PYCR1) is constitutively localized in the nucleus of colorectal cancer cells. In hypoxia conditions, IGF1R phosphorylates PYCR1 at Tyrosine135 and promotes the binding of PYCR1 with ETS-like transcription factor 4 (ELK4). Mechanistically, this complex is key to suppressing the expression of genes associated with cell growth arrest. Specifically, the PYCR1 reaction generates NAD⁺, which supports the deacetylase activity of Sirt7 and, in turn, gene transcription suppression by ELK4. Notably, the authors were able to associate this specific PYCR1 phosphorylation with colorectal cancer growth.

Overall, the manuscript presents an interesting and novel mechanism for the PYCR1 role and its capacity to promote cancer cell growth under hypoxia. The biochemistry studies include the proper controls and thoroughly show the mechanistic link between IGF1R, PYCR1, Sirt7, and ELK4. Moreover, the therapeutic link is also present. In conclusion, I encourage the publication of this study in Nature Communications. I only have a few comments that I believe will strengthen even more the main conclusions.

Answer: We greatly appreciate the reviewer's acknowledgement of the significance of the present study and the insightful comments that will definitely strengthen our manuscript.

Major:

1. In Figure 1B, the authors show that PYCR1 is localized in the nucleus also in normoxia. Does nuclear PYCR1 interact with other genes in normoxia? A better characterization of PYCR1 interaction in these conditions will help strengthen its specific role in hypoxia. The mechanistic characterization of PYCR1 interaction in normoxia is not required.

Answer: Thanks for pointing out the issue. In the revised manuscript, the interaction of PYCR1 with BIP as revealed from mass spectrometry (Supplementary Fig. 1d) was further confirmed (Supplementary Fig. 1f), the intensity of which was no sound difference between cells culturing under normoxia and hypoxia, reflecting the specific effect of hypoxia on the PYCR1-ELK4 interaction.

2. All the in vitro studies of this manuscript were performed in 1% oxygen. Do the authors observe a different response of tumor cells in vivo depending on the region in which they are localized (normoxic regions vs hypoxic regions)?

Answer: Thanks for pointing out this important issue. According to reviewer's suggestion, we analyzed nuclear level and phosphorylation level of PYCR1 and in clinical tumor specimens and distinct regions of xenografted tumors lysates. As a result, the intensity of PYCR1 pY135 (Supplementary Fig. 6g) rather than the total level of PYCR1 (Supplementary Fig. 1a) correlates Hif-1 \$\alpha\$ levels collected clinical tumor specimens. Meanwhile, in distinct regions of xenografted tumors with higher level of HIF-1 \$\alpha\$ levels, PYCR1 pY135 was generally detected with a stronger signals, while which was not occurred in the total protein level of PYCR1 (Supplementary Fig.

6e). These data suggest nuclear localization of PYCR1 is localized in the nucleus regardless of intratumor oxygen condition, and PYCR1 phosphorylation is induced under hypoxia.

3. P5CS is normally located in the mitochondria. Does the product of the P5CS reaction diffuse to the cytosol and then to the nucleus? Did the authors find P5CS in the nucleus? More experiments elucidating the compartmentalization of proline biosynthesis are needed.

Answer: Thanks for pointing out this important issue and we performed new experiment according to reviewer's suggestion. As a result, P5CS was found to be localized in the nucleus, and its level (Supplementary Fig. 5h) as well as the metabolic activity (Supplementary Fig. 5i) was even slightly increased under hypoxia. In the meantime, metabolite of nuclear fraction (Please see in the section "Methods") further showed the nuclear amount of proline was increased under hypoxia (Supplementary Fig. 5f), which was largely decreased by P5CS depletion. Similar to the overall level of proline level (Fig.2j), the nuclear level of proline was augmented under hypoxia in WT rPYCR1- or rPYCR1 Y135F-expressing cells (Supplementary Fig. 2o), which was blocked in rPYCR1 T238A-expressing cells. These data reveal an enhancement of the sequential reactions for proline synthesis at this condition. In terms of measurement of P5C amount, the pretest from mass spectrometry platforms we tried to contact revealed it is unable to stably and precisely proceed the experiment, which would be due to the chemical properties of P5C. Here we hope this issue could be understood and believe the technical improvement in this regard will be definitely beneficial for the further study.

Moreover, results shown in Fig. 5f indicated expression of addition of combined P5C and NADH facilitated promoter-associated H3K18 deacetylation in vitro, while this effect was blocked by SIRT7 silencing, suggesting the essential role of SIRT7. The in vitro assay also showed indicated addition NAD⁺ but not P5C-NADH incubation was able to remove promoter-associated H3K18 deacetylation in PYCR1-depleted cells (Supplementary Fig. 5j), which was inhibited by SIRT7 silencing. These data further demonstrate the activity of PYCR1 for NAD⁺ production is required for SIRT7-mediated H3K18 deacetylation at promoter of ELK4-targeted genes.

Minor:

1. Some of the sentences in the manuscript are oddly articulated and thus difficult to understand. I suggest revising the manuscript so that it is easier to follow by the readers.

Answer: Thanks for pointing out the issue. In the revised manuscript, we have made the according modification to improve the writing quality that would facilitate the reading.

REVIEWERS' COMMENTS

Reviewer #1 (Remarks to the Author):

The authors addressed all of my comments. I think this is a very interesting and important finding and should be shared among the scientific community by publishing it in Nat Commun.

Reviewer #2 (Remarks to the Author):

In the revised version of the manuscript, the authors have done an impressive amount of work and have addressed satisfactorily all my concerns. The new evidence strengthens considerably the model proposed by the authors regarding the PYCR1-ELK4-SIRT7 regulatory axis under hypoxia. I only have a minor issue. Some experiments lack statistical analysis, which is a standard requirement for these quantifications. This is the case for Figures 1h, 2j and Suppl figures 20, 5f and 5i.

Reviewer #3 (Remarks to the Author):

The authors have sufficiently answered all my comments. The manuscript now presents additional experiments that further strengthen the author's conclusions. Thus, I encourage the publication of this study in Nature Communications.

Reviewer #1 (Remarks to the Author):

The authors addressed all of my comments.

I think this is a very interesting and important finding and should be shared among the scientific community by publishing it in Nat Commun.

Answer: We greatly appreciate the reviewer's acknowledgement of the significance of the present study and the professional suggestions.

Reviewer #2 (Remarks to the Author):

In the revised version of the manuscript, the authors have done an impressive amount of work and have addressed satisfactorily all my concerns. The new evidence strengthens considerably the model proposed by the authors regarding the PYCR1-ELK4-SIRT7 regulatory axis under hypoxia. I only have a minor issue. Some experiments lack statistical analysis, which is a standard requirement for these quantifications. This is the case for Figures 1h, 2j and Suppl figures 20, 5f and 5i.

Answer: We greatly appreciate the reviewer's professional suggestions. We have revised the manuscript accordingly.

Reviewer #3 (Remarks to the Author):

The authors have sufficiently answered all my comments. The manuscript now presents additional experiments that further strengthen the author's conclusions. Thus, I encourage the publication of this study in Nature Communications.

Answer: We greatly appreciate the reviewer's acknowledgement of the significance of the present study and the professional suggestions.